# The TDRD3-USP9X complex and MIB1 regulate TOP3B homeostasis and prevent deleterious TOP3B cleavage complexes

Sourav Saha[1], Shar-yin Naomi Huang[1], Xi Yang [1], Liton Kumar Saha [1], Yilun Sun [1], Prashant Khandagale[1], Lisa M. Jenkins [2] & Yves Pommier [1] ✉

TOP3B is stabilized by TDRD3. Hypothesizing that TDRD3 recruits a deubiquitinase, we find that TOP3B interacts with USP9X via TDRD3. Inactivation of USP9X destabilizes TOP3B, and depletion of both TDRD3 and USP9X does not promote further TOP3B ubiquitylation. Additionally, we observe that MIB1 mediates the ubiquitylation and proteasomal degradation of TOP3B by directly interacting with TOP3B independently of TDRD3. Combined depletion of USP9X, TDRD3 and MIB1 causes no additional increase in TOP3B levels compared to MIB1 knockdown alone indicating that the TDRD3-USP9X complex works downstream of MIB1. To comprehend why cells degrade TOP3B in the absence of TDRD3, we measured TOP3Bccs. Lack of TDRD3 increases TOP3Bccs in DNA and RNA, and induced R-loops, γH2AX and growth defect. Biochemical experiments confirm that TDRD3 increases the turnover of TOP3B. Our work provides molecular insights into the mechanisms by which TDRD3 protect cells from deleterious TOP3Bccs which are otherwise removed by TRIM41.

Topoisomerases are vital cellular enzymes preventing and resolving nucleic acid topological problems during replication, transcription, recombination, chromosome segregation and chromatin remodeling[1–3]. Although vertebrates have six topoisomerases (TOP1, TOP1mt, TOP2A, TOP2B, TOP3A, and TOP3B), only topoisomerase 3β (TOP3B) can catalytically work on DNA and RNA both in vitro and in vivo and participates in DNA and RNA metabolic processes[4–12].

TOP3B is important for maintaining genome stability as it decreases cellular R-loops by relaxing hypernegative supercoil or by resolving R-loops in coordination with the helicase DDX5[10,12–16]. The R-loop suppressing activity of TOP3B also helps during transcription in cancer cells and neuronal activity-dependent transcription in mouse brain tissue[9,10]. In parallel, TOP3B is associated with polyribosomes and translating mRNA pools, localizes to RNA stress granules (cytoplasmic membrane-less assemblies of non-translating mRNAs and proteins formed in response to cellular stress) and promotes translation of subsets of RNAs in cancer cells and neurons[5–8,11,17]. TOP3B binding also stabilizes a subset of RNAs via a topoisomerase-independent

mechanism in HCT116 colon carcinoma cells[5]. TOP3B malfunctions are associated with breast cancer[18], bilateral renal cancer[15], schizophrenia, and autism spectrum disorders[6–8,11,17].

In cells, TOP3B works as a heterotrimeric complex with its auxiliary factor TDRD3 (Tudor domain-containing protein 3) and FRMP (Fragile X Mental Retardation Protein) by forming the TOP3B-TDRD3-FMRP complex (TTF complex)[6,7]. TOP3B directly interacts with TDRD3 at least via its Domain II and the DUF-OB fold of TDRD3, and with the 'Tudor domain' of TDRD3, which recognizes dimethylated arginine residues (Arg833me2 and Arg835me2) in the C-terminal RGG domain of TOP3B[6,7,10,13,19]. FMRP associates with the TTF complex via TDRD3's C-terminal FMRP Interacting Motif (FIM) but without directly interacting with TOP3B[7].

TDRD3 is a multidomain and multifunctional scaffolding protein. Its 'Tudor domain' "reads" asymmetric dimethylarginine (ADMA) marks in the RG-rich domains of different proteins including not only TOP3B, but also RNA pol II and histones[10,13,20–23]. TDRD3 binds single-stranded DNA and RNA[24]. TDRD3 has been shown to play a pivotal role

[1]Developmental Therapeutics Branch & Laboratory of Molecular Pharmacology, Center for Cancer Research, National Cancer Institute, NIH, Bethesda, MD 20892, USA. [2]Collaborative Protein Technology Resource, Center for Cancer Research, National Cancer Institute, NIH, Bethesda, MD 20892, USA. ✉e-mail: pommier@nih.gov

in orchestrating the activities of TOP3B by directly stimulating the DNA and RNA topoisomerase activities of TOP3B[24,25] and targeting TOP3B to chromatin (via its Tudor domain)[10] and RNA (via its DUF-OB and exon junction complex binding motif)[7]. *Drosophila* TDRD3 also recruits TOP3B to the siRNA machinery (RISC complex containing p68 and AGO2) to promote heterochromatin formation and transposon silencing[26]. Another important cellular function of TDRD3 is to stabilize TOP3B and in the absence of TDRD3, TOP3B has been reported to be ubiquitylated and degraded by the proteasome[5,10].

In the present study, we aimed to answer two fundamental questions: (1) How does TDRD3 limit the ubiquitylation and subsequent degradation of cellular TOP3B? and (2) Why do cells need to degrade TOP3B when they lack TDRD3? To address these questions, we carried out cellular and pharmacological experiments that establish that TDRD3 protects TOP3B by recruiting the deubiquitinase (DUB) USP9X. We also find that USP9X antagonizes TOP3B ubiquitylation by the E3 ligase Mind bomb E3 Ubiquitin Protein Ligase 1 (MIB1). To elucidate why cells degrade TOP3B in the absence of TDRD3, we provide cellular and biochemical evidence that, in the absence of TDRD3, cellular DNA/ RNA TOP3B cleavage complexes (TOP3Bccs) accumulate as abortive and cytotoxic intermediates.

## Results

### TDRD3 limits TOP3B ubiquitylation and stabilizes TOP3B in human cells

First, to elucidate how TDRD3 stabilizes TOP3B, we checked TOP3B protein levels in human embryonic kidney HEK293 cells transfected with siRNA against TDRD3. TOP3B protein levels went down in parallel with TDRD3 protein levels (Fig. 1a, b). We observed a similar reduction in TOP3B protein levels in HCT116 colon cancer cell line transfected with siTDRD3 (Fig. 1c) and in *TDRD3KO* HCT116 (in comparison to wild-type cell; Fig. 1d)[5]. To verify the role of TDRD3 in preventing the proteolysis of TOP3B, we performed cycloheximide (CHX; 10 µg/ml) chase experiments and western blotting in TDRD3 depleted conditions i.e., in siTDRD3-transfected HEK293 cells and in *TDRD3KO* HCT116 cells. TOP3B displayed a shorter half-life both in siTDRD3-transfected HEK293 cells (in comparison to control transfected cells; Fig. 1e, g) and in *TDRD3KO* HCT116 cells (in comparison to wild-type cells; Fig. 1f, h). Conversely, ectopic expression of TDRD3 stabilized TOP3B both in HEK293 and *TDRD3KO* HCT116 cells (Fig. 1i–l). These experiments demonstrate the role of TDRD3 in stabilizing TOP3B in two different cellular systems[10].

We also checked TOP3B mRNA levels in TDRD3 depleted HEK293 and *TDRD3KO* HCT116 cells to confirm previous observation in MCF7 cell line that loss of TDRD3 does not affect TOP3B transcript levels[10]. In both the cell lines TDRD3 downregulation did not lower TOP3B RNA level in cells (Supplementary Fig. 1a, b).

Because TOP3B has been reported to be ubiquitylated and targeted for proteasomal degradation in MCF7 breast cancer cells in the absence of TDRD3[10], we measured the ubiquitylation of TOP3B in TDRD3-depleted cells (i.e., in siTDRD3-transfected HEK293 cells and in *TDRD3KO* HCT116 cells). To detect ubiquitylation, cells were transiently transfected with HA-tagged ubiquitin. Immunoprecipitation (IP) with anti-TOP3B antibody and subsequent Western blotting showed elevated levels of ubiquitylated TOP3B in TDRD3 depleted conditions (Fig. 2a, b). Conversely, ectopic expression of TDRD3 in *TDRD3KO* HCT116 cells reduced TOP3B ubiquitylation (Fig. 2c). These results generalize the prior findings[10] that TDRD3 limits TOP3B ubiquitylation and thereby stabilizes TOP3B in cells.

To determine the ubiquitin linkages responsible for polyubiquitylation of free TOP3B in the absence of TDRD3 we transfected *TDRD3KO* HCT116 cells with either wild-type HA-tagged ubiquitin (Ub) or HA-tagged lysine-to-arginine Ub mutants for each of those 7 lysine residues (K6R-Ub, K11R-Ub, K27R-Ub, K29R-Ub, K33R-Ub, K48R-Ub, and K63R-Ub), pulled down TOP3B and probed with ubiquitin

antibody. Results indicate that both K11 and K48-linked ubiquitin chains are critical for TOP3B polyubiquitylation (Supplementary Fig. 1c), which is consistent with the proteasomal degradation of free TOP3B in the absence of TDRD3.

### The deubiquitinase USP9X stabilizes TOP3B bound to TDRD3

To explain the increased ubiquitylation of TOP3B in the absence of TDRD3, we hypothesized that TDRD3 recruits a DUB to TOP3B. To search for TOP3B-interacting DUBs, we immunoprecipitated endogenous TOP3B and performed liquid chromatography-tandem mass spectrometry (LC-MS/MS) in HEK293 cells (IP-MS experiment reported previously[12]). Endogenous TOP3B pulldown in HEK293 cells retrieved 755 significantly enriched proteins (with peptide-spectrum match [PSM] value > 10), including, as expected TDRD3 (Fig. 2d). Several DUBs, USP9X, USP39, USP7, USP10 were found as interactors of human TOP3B (Fig. 2d). Among which, USP9X displayed the highest PSM value. To confirm these results, we repeated the IP-LC-MS/MS analysis in HEK293 cells transfected with hemagglutinin (HA)-tagged TOP3B (IP-MS experiment reported previously in[12]). Pulling down HA-TOP3B retrieved 471 proteins with PSM values > 10. Again, we found USP9X as a top interactor of TOP3B (Fig. 2e).

To validate the TOP3B-USP9X interaction discovered by IP-MS, we performed reciprocal IP with USP9X antibody followed by Western blotting both in HEK293 and HCT116 cells. Consistent with the IP-MS results, USP9X was detected in the immunoprecipitates of TOP3B (Fig. 2f–h). To determine whether the TOP3B-USP9X interaction is TDRD3-dependent, we performed USP9X IP-Western blot experiments in TDRD3-depleted conditions, i.e. in siTDRD3 transfected HEK293 cells and in *TDRD3KO* HCT116 cells. MG132 was added to prevent the proteasomal degradation of TOP3B in the absence of TDRD3. The USP9X-TOP3B interaction was significantly reduced in siTDRD3-transfected HCT116 cells and no TOP3B-USP9X interaction was detected in *TDRD3KO* HCT116 cells (Fig. 2f, g). These results demonstrate that USP9X interacts with TOP3B only in the presence of TDRD3.

An earlier study showed that C-terminal of TDRD3 interacts with USP9X's C-terminal domain[27]. To show that TDRD3 cannot stabilize TOP3B in the absence of the USP9X interaction motif, we generated a C-terminal deletion mutant of TDRD3 (1-352 TDRD3-FLAG), which we overexpressed in *TDRD3KO* HCT116 cells in comparison with TDRD3 wild- type counterpart (Schematic representations of the domain organization of wild-type TDRD3 and C-terminal deletion mutant of TDRD3 are shown in Supplementary Fig. 1d). C-terminal deletion of TDRD3 (1-352 aa) could not stabilize TOP3B protein compared to wild-type TDRD3 protein (Supplementary Fig. 1e). These results indicate that USP9X deubiquitylates TOP3B only when bound to TDRD3.

### USP9X deubiquitylates and stabilizes TOP3B in the presence of TDRD3

To determine the importance of USP9X for the cellular stability of TOP3B, we transfected HEK293 cells with siRNAs against USP9X. Depletion of USP9X reduced the cellular levels of TOP3B protein (Fig. 3a). USP9X downregulation also lowered TDRD3 levels[27]. To confirm that USP9X stabilizes TOP3B and TDRD3, we performed cycloheximide (CHX; 10 µg/ml) chase experiment in USP9X-depleted conditions. Knocking down USP9X markedly decreased the half-lives of both TOP3B and TDRD3 proteins (Fig. 3b, c).

To further demonstrate that USP9X stabilizes TOP3B, we treated HEK293 cells with the USP9X inhibitor Degrasyn [WP1130, (2 µM, 24 h)]. As positive controls, cells were either treated with the pan-DUB inhibitor PR-619 (5 µM, 24 h) or transfected with siRNAs against TDRD3 (48 h). Similar to PR-619 treatment or TDRD3 down-regulation, pharmacological inhibition of USP9X by WP1130 reduced TOP3B levels (Fig. 3d), which is consistent with our USP9X down-regulation

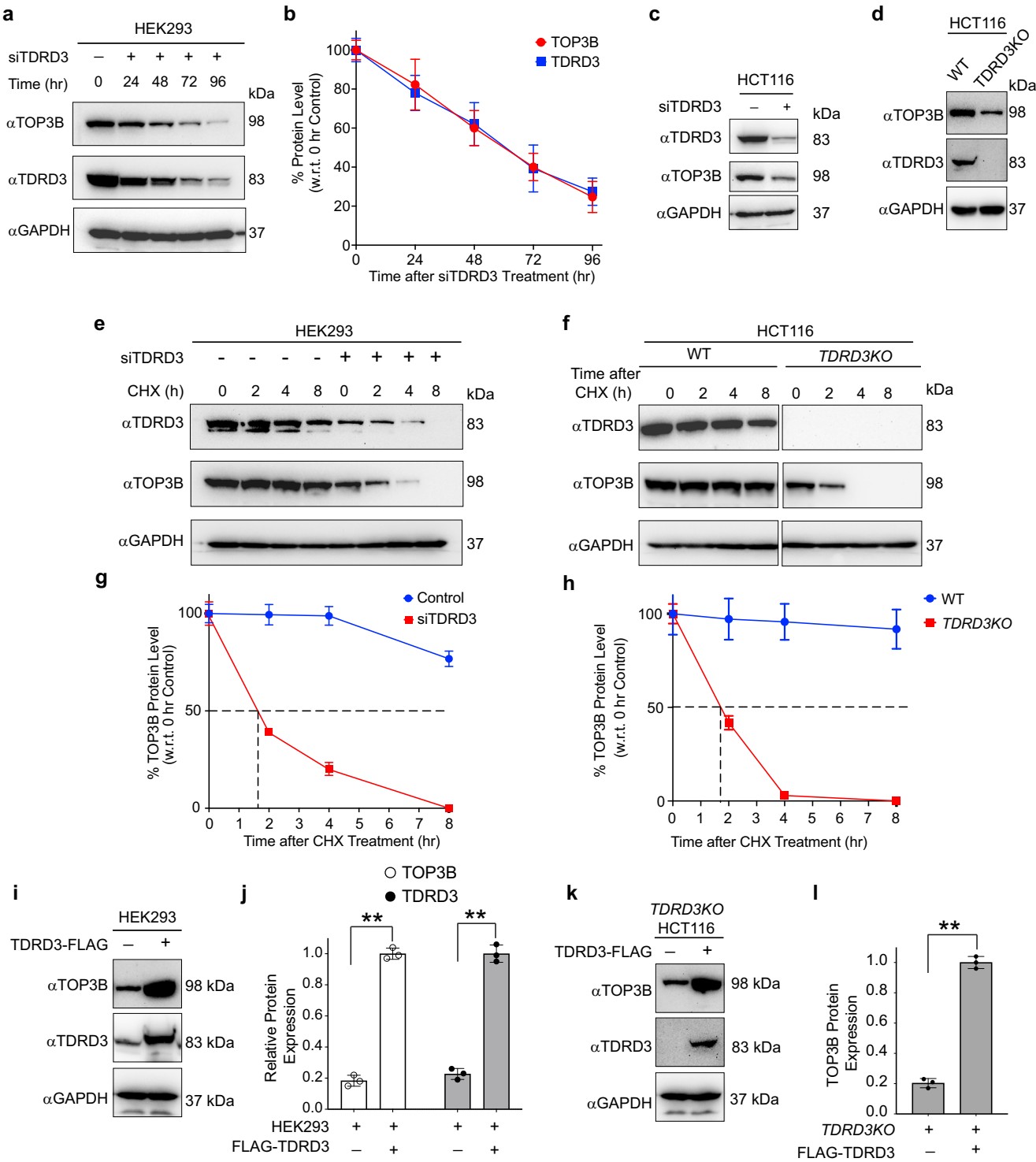

experiment (Fig. 3a–c). PR-619 treatment or WP1130 treatment also lowered TDRD3 levels (Fig. 3d). These results demonstrate that USP9X determines the stability of both TOP3B and TDRD3 in cells.

To establish that USP9X acts as a DUB for TOP3B, we carried out TOP3B pulldown experiment and measured the ubiquitylation of TOP3B after cellular depletion of USP9X in cells transiently transfected with HA-tagged ubiquitin. As control, we also downregulated TDRD3, which increased ubiquitylated TOP3B (Fig. 3e)[10]. Downregulation of USP9X markedly increased TOP3B ubiquitylation (Fig. 3e), and depletion of both USP9X and TDRD3 produced no further increase in TOP3B ubiquitylation compared to USP9X- or TDRD3-depletion (Fig. 3e). Combined, these results (Figs. 3e and 2f, g) show that USP9X acts as a

DUB for TOP3B and that TDRD3 recruits USP9X to TOP3B to stabilize TOP3B by deubiquitylating TOP3B in cells.

To establish whether USP9X-mediated deubiquitylation protects TOP3B from proteasomal degradation, we treated siUSP9X-transfected cells with the proteasome inhibitor MG132 (1 μM, 24 h). Treatment with MG132 restored both TOP3B and TDRD3 protein levels in USP9X-depleted cells (Fig. 3f) consistent with the role of USP9X in protecting both TOP3B and TDRD3 from their proteasomal degradation in cells. To confirm the role of TDRD3 in mediating TOP3B deubiquitylation by USP9X, we depleted USP9X in *TDRD3KO* HCT116 cells. As anticipated, downregulation of USP9X did not affect TOP3B stability in *TDRD3KO* HCT116 cells (Fig. 3g), which further establishes the

**Fig. 1 | TDRD3 maintains TOP3B protein levels in HEK293 and HCT116 cells.**
**a**, **b** TDRD3 stabilizes cellular TOP3B protein in HEK293 cells. Cell lysates from
TDRD3 siRNA transfected cells were immunoblotted with TOP3B and TDRD3
antibodies (GAPDH as loading control). Panel **a** is a representative blot and panel
**b** the quantitation of TOP3B and TDRD3 protein levels from three independent
experiments. Data are plotted as means ± standard deviations (SD). **c** TDRD3
depletion by siRNA destabilizes TOP3B protein in HCT116 cells. **d** TOP3B expression
levels in HCT116 wild-type and *TDRD3KO* cells. **e**, **f** Depletion of TDRD3 decreases
the half-life of TOP3B. HEK293 cells were transfected with TDRD3 siRNA and treated
with CHX (10 μg/mL) for the indicated time. Panel **e** is a representative western blot
and panel **f** the quantitation of three independent experiments. Data are plotted as
means ± standard deviations (SD). **g**, **h** TDRD3 increases half-life of TOP3B protein
in HCT116 cells. HCT116 wild-type and *TDRD3KO* cells were treated with CHX (10 μg/
mL) for the indicated time. Panel **g** is a representative western blot and panel

**h** displays quantitation of TOP3B protein levels from three independent experiments. Data are plotted as means ± standard deviations (SD). **i**, **j** Ectopic expression
of TDRD3 stabilizes TOP3B in HEK293 cells. Cells were transfected with TDRD3-
FLAG constructs. Panel **i** is a representative western blot and panel **j** the quantitation of three independent experiments. Data are plotted as means ± standard
deviations (SD). Two-tailed paired *t*-test. \*\**P* value = 0.0003 (TOP3B in no transfected vs TDRD3 transfected), \*\**P* value = 0.0002 (TDRD3 in no transfected vs
TDRD3 transfected). **k**, **l** Similar experiment in TDRD3 knockout HCT116 cells.
Ectopic expression of TDRD3 stabilizes TOP3B in *TDRD3KO* HCT116 cells. *TDRD3KO*
HCT116 cells were transfected with TDRD3-FLAG constructs. Panel **k** is a representative western blot and panel **l** displays quantitation of TOP3B protein level from
three independent experiments. Data are plotted as means ± standard deviations
(SD). Two tailed unpaired t-test. \*\**P* value = <0.0001. Source data are provided as a
Source Data file.

importance of the TDRD3-USP9X interaction in stabilizing
cellular TOP3B.

To determine whether USP9X can work on the proteasomal targeting K11- and K48-linked ubiquitin chains formed on free TOP3B, we
pulled down TOP3B from HEK293 cells after siRNA-mediated USP9X
downregulation or after ectopic expression of USP9X-FLAG and probed with K11- and K48-specific ubiquitin antibodies. Results indicated
that USP9X can remove both K11- and K48-linked ubiquitin chains from
free TOP3B (Supplementary Fig. 2a, b).

We also generated a catalytically inactive USP9X (USP9X C1566S
mutant having no DUB activity), which we over-expressed in HEK293
cells in comparison with its wild-type counterpart. Overexpression of
wild-type USP9X stabilized TOP3B and TDRD3 whereas the USP9X
C1566S mutant failed to stabilize both the proteins (Fig. 3h and Supplementary Fig. 2c). This result confirms that the DUB activity of
USP9X is responsible for the stabilization of TOP3B and TDRD3.

Altogether, we conclude that USP9X deubiquitylates and stabilizes TOP3B and that TDRD3 acts by mediating the deubiquitylation of
TOP3B by USP9X.

## Mind bomb E3 Ubiquitin Protein Ligase 1 ubiquitylates cellular TOP3B independently of TDRD3

For our results thus far established that TOP3B is ubiquitylated and
targeted for proteasomal degradation unless it is deubiquitylated and
stabilized by the TDRD3-USP9X complex, we sought out to identify the
E3 ubiquitin ligase responsible for the ubiquitylation and proteasomal
targeting of free TOP3B. We hypothesized that MIB1 (Mind bomb 1),
which belongs to the RING domain-containing family of E3 ubiquitin
ligases, was a plausible candidate as it has been reported to interact
with USP9X, and MIB1-USP9X often works as a counteracting pair to
maintain homeostatic level of different cellular proteins such as
PCM1[28–31]. Furthermore, we found MIB1 in prior pull-down studies of
TOP3B[32,33] and TOP3B, TDRD3, USP9X has been proposed to be part of
a large macromolecular complex containing MIB1[32].

We demonstrated the TOP3B-MIB1 interaction by IP-western
blotting with both TOP3B and MIB1 antibodies in HEK293 cells. Indeed,
TOP3B pulled down MIB1 (Fig. 4a) and vice-versa, MIB1 pulled down
TOP3B (Fig. 4b). IP-Western blot experiments using TDRD3 antibody
also showed MIB1 interaction with TDRD3 in HEK293 cells (Fig. 4c).

To determine whether the TOP3B-MIB1 interaction is TDRD3-
dependent, we performed TOP3B IP-Western blotting in TDRD3-
depleted condition, i.e., in siTDRD3-transfected HEK293 cells with
MG132 added to prevent the proteasomal degradation of TOP3B.
Although TDRD3 was found to interact both with TOP3B (Fig. 4d)[6,7,10,12]
and MIB1 (Fig. 4d), depletion of TDRD3 did not affect the TOP3B-MIB1
interaction (Fig. 4d). These results indicate that MIB1 is recruited to
TOP3B independently of TDRD3.

Next, we tested whether MIB1 acts as E3 ubiquitin ligase of TOP3B.
By transfecting HEK293 cells with siMIB1, we found increased accumulation of TOP3B protein compared to control siRNA-transfected

cells (Fig. 4e), while siMIB1 was without effect on cellular TDRD3 levels
(Fig. 4e). We also carried out TOP3B pull-down experiments and
examined the ubiquitylation status of TOP3B after cellular depletion of
MIB1 or TDRD3, and after combined depletion of MIB1 and TDRD3
(Fig. 4f). While TDRD3 depletion increased ubiquitylated TOP3B species, down-regulation of MIB1 reduced the ubiquitylation of TOP3B
(Fig. 4f), and depletion of both MIB1 and TDRD3 produced no further
decrease in TOP3B ubiquitylation compared to MIB1-depleted cells
(Fig. 4f). As K11- and K48-linked ubiquitin chains are responsible for
polyubiquitylation of TOP3B in the absence of TDRD3 (see Supplementary Fig. 1c), we determined whether MIB1 can form K11- and K4-
linked ubiquitin chains on TOP3B. Pulling-down TOP3B from *TDRD3KO*
HCT116 cells after siRNA-mediated MIB1 downregulation or after
ectopic expression of MIB1-FLAG and probing with K11- and K48-
specific ubiquitin antibodies indicated that MIB1 forms both K11- and
K48-linked-ubiquitin chains from free TOP3B (Supplementary
Fig. 3a, b).

To further test the implication of E3 ligase activity of MIB1 in the
ubiquitylation and proteasomal degradation of TOP3B, we over-
expressed catalytically inactive MIB1 (MIB1 C985S active site mutant
having no E3 ligase activity) in comparison with its wild-type counterpart (Fig. 4g and Supplementary Fig. 3c). Overexpression of wild-
type MIB1 increased ubiquitylation of TOP3B, decreased total cellular
TOP3B level whereas MIB1 C985S mutant failed to affect TOP3B protein ubiquitylation or stability (Fig. 4g and Supplementary Fig. 3c).
Combined, these results show that the E3 ubiquitin ligase activity of
MIB1 is required for the ubiquitylation and proteasomal
degradation TOP3B.

As MIB1 can also interact with TDRD3 (Fig. 4c), we also checked
the ubiquitylation of TDRD3 after depletion of MIB1. In contrast to
TOP3B (Fig. 4f), *MIB1* down-regulation did not affect TDRD3 ubiquitylation (Fig. 4h), indicating that MIB1 is not an E3 ubiquitin
ligase for TDRD3. From these results, we conclude that MIB1 acts
upstream of TDRD3 as an ubiquitin ligase for free TOP3B independently of TDRD3.

## The TDRD3-USP9X complex regulates TOP3B homeostasis downstream from MIB1

To elucidate the relationship between the TDRD3-USP9X complex
and MIB1, we measured cellular TOP3B levels after down-
regulating TDRD3, USP9X, and MIB1 alone or in a pairwise manner, i.e., by downregulating TDRD3-USP9X, TDRD3-MIB1, and
USP9X-MIB1. Consistent with our above results, TDRD3 and
USP9X downregulations caused depletion of cellular TOP3B as
well as of TDRD3 (Fig. 4i) whereas MIB1 downregulation only
caused an accumulation of TOP3B without affecting TDRD3 levels
(Fig. 4i). Combined depletion of TDRD3-USP9X did not cause any
further change in TOP3B levels compared to only TDRD3 or
USP9X depleted conditions (Fig. 4i). Finally, MIB1 downregulation
restored the TOP3B protein levels in both TDRD3 and USP9X

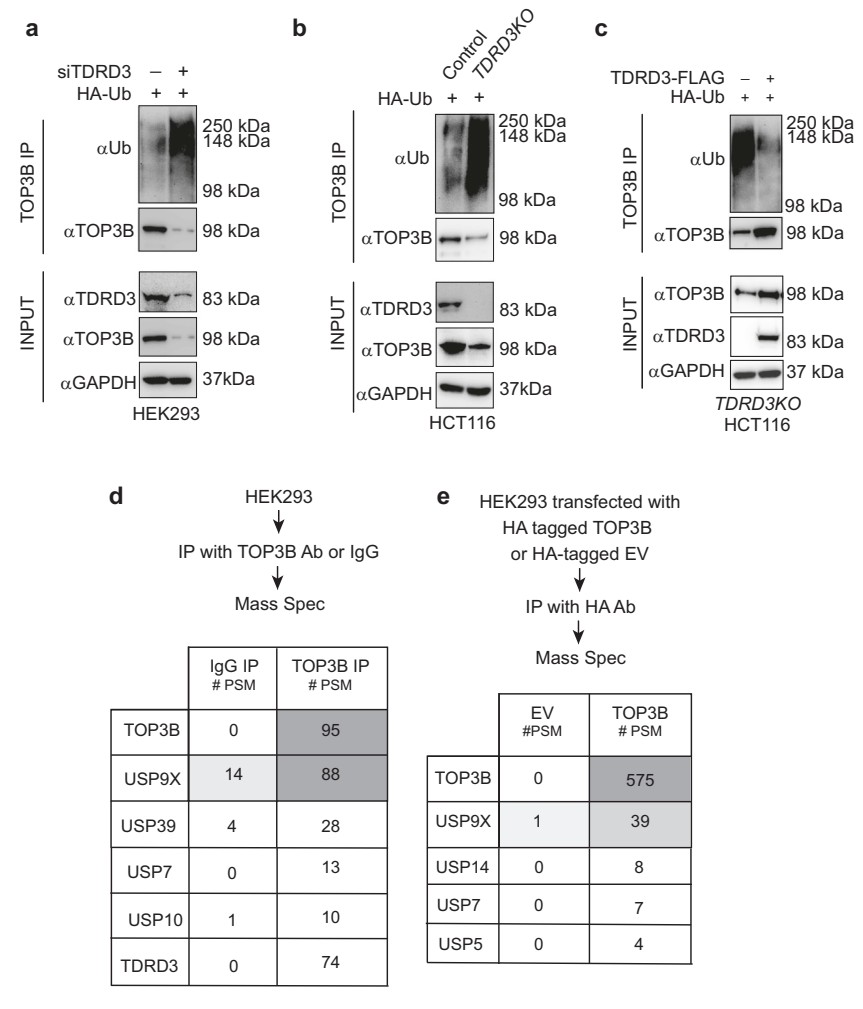

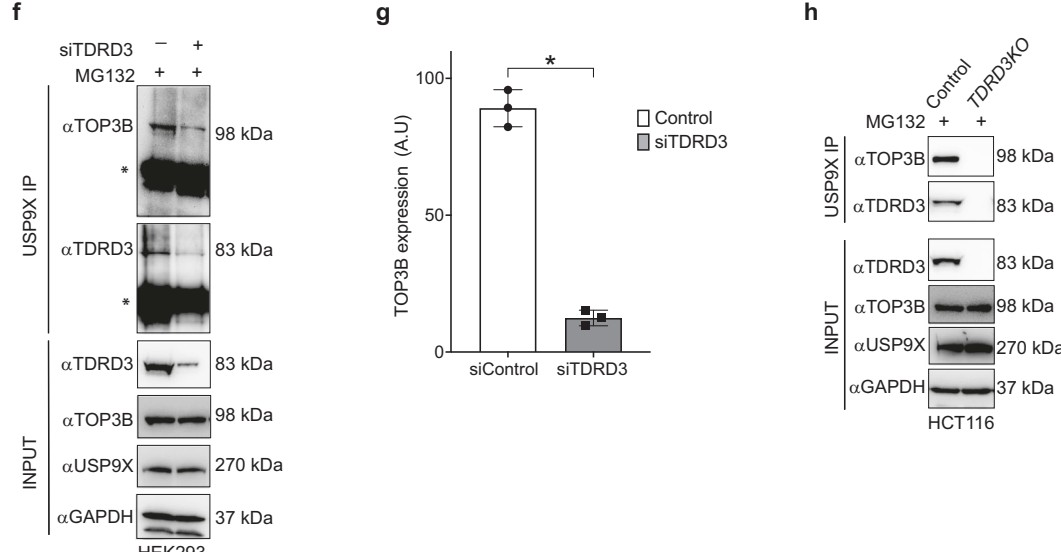

depleted conditions (Fig. 4i) without affecting TDRD3 (Fig. 4i). Also, combined depletion of USP9X-MIB1 or TDRD3-MIB1 caused no further increase in TOP3B levels compared to MIB1 knockdown alone (Fig. 4i). From these observations, we conclude that TDRD3-USP9X acts downstream of MIB1 and antagonizes the action of MIB1, which acts as an E3 ubiquitin ligase of TOP3B, and that MIB1 ubiquitylates and destabilizes TOP3B by proteasomal targeting

whereas TDRD3-USP9X stabilizes TOP3B by removing MIB1-mediated ubiquitylation (Fig. 4j).

## TDRD3 protects cells from abortive TOP3B cleavage complexes and TOP3Bcc-induced genomic damage

As irreversible/abortive topoisomerase cleavage complexes (TOPccs) can cause cellular damage[1,4,34,35], we hypothesized that cells degrade

**Fig. 2 | Enhanced ubiquitylation of TOP3B in the absence of TDRD3 and recruitment of USP9X by TDRD3. a** Enhanced TOP3B ubiquitylation in the absence of TDRD3 in HEK293 cells. Cells were transfected with HA-Ubiquitin construct and TDRD3 siRNA (as indicated) for 48 h and subjected to TOP3B pulldown. Pulled down and input samples were resolved on SDS-PAGE and probed with Ubiquitin, TOP3B and TDRD3 antibodies. The figure is representative of three independent experiments. **b** Enhanced ubiquitylation of TOP3B in *TDRD3KO* HCT116 cells. HCT116 wild-type and *TDRD3KO* cells were subjected to TOP3B pulldown. Pulled down and input samples were resolved on SDS-PAGE and probed with Ubiquitin, TOP3B and TDRD3 antibodies. The figure is representative of three independent experiments. **c** Ectopic expression of TDRD3 decreases TOP3B ubiquitylation in *TDRD3KO* HCT116 cells. *TDRD3KO* HCT116 cells were transfected with TDRD3-FLAG constructs for 48 h and subjected to TOP3B pull down. Pulled down and input samples were resolved on SDS-PAGE and probed with Ubiquitin, TOP3B

and TDRD3 antibodies. The figure is representative of three independent experiments. **d** TOP3B pulldown-LC-MS/MS showing that endogenous TOP3B interacts with USP9X and TDRD3 in HEK293 cells. Shown are the number of identified peptide spectral matches (PSM). **e** HA-tag pulldown-LC-MS/MS showing TOP3B interaction with USP9X in HEK293 cells. After transfection of HA-TOP3B, HEK293 cells were subjected to HA-tag pulldown followed by LC-MS. **f, g** USP9X pulldown-Western blot experiments showing that depletion of TDRD3 suppresses TOP3B-USP9X interaction in HEK293 cells. Panel **f** is a representative Western blot and panel **g** is the quantitation of three independent experiments. Data are plotted as means ± standard deviations (SD). Two tailed unpaired *t*-test. *P value = 0.002. **h** USP9X pulldown-Western blot experiments showing the absence of TOP3B-USP9X interaction in *TDRD3KO* HCT116 cells. Source data are provided as a Source Data file.

TOP3B in the absence of TDRD3 to avoid unregulated cellular DNA/RNA cleavage complexes and subsequent genomic damage. To measure TOP3Bccs in the absence and presence of TDRD3, we used *TDRD3KO* HCT116 cells[5] transfected with HA-TOP3B construct alone or in combination with FLAG-tagged TDRD3 construct (Fig. 5a), and measured TOP3Bccs after IP enrichment of RADAR assay samples using HA antibody[4,12]. We overexpressed TDRD3 to maintain the TOP3B:TDRD3 stoichiometry in *TDRD3KO* HCT116 cells after ectopic expression of TOP3B. Overexpression of TOP3B and TDRD3 were confirmed by Western blotting, and as expected, absence of TDRD3 reduced the levels of cellular HA-TOP3B protein (Fig. 5a). Unexpectedly, RADAR assays revealed that cells also accumulate more TOP3Bccs in the absence of TDRD3 (Fig. 5b, c).

To study the effect of TDRD3 depletion on endogenous TOP3Bcc, we performed RADAR assays to try and detect endogenous TOP3Bcc and could not detect endogenous TOP3Bcc in TDRD3 depleted HCT116 cells even after MG132 treatment (Supplementary Fig. 4a, b). So we used recently reported TOP3B trapping drugs NSC96932 and NSC690634[36]. Wild-type and *TDRD3KO* HCT116 cells were treated with MG132 (10 μM, 3 h) and NSC96932 (100 μM, 1 h) or NSC690634 (100 μM, 1 h) and RADAR assays were performed to detect endogenous TOP3Bcc. *TDRD3KO* cells accumulated more endogenous TOP3Bcc compared to their wild-type counterpart (Supplementary Fig. 4a, b). This finding corroborates our previous results using *TDRD3KO* HCT116 cells transfected with HA-TOP3B construct alone or in combination with FLAG-tagged TDRD3 construct (Fig. 5a). They imply that TDRD3 controls the activity of TOP3B by limiting the accumulation of TOP3Bcc catalytic intermediates.

In our earlier work, we found that endogenous TOP3Bccs could be detected following IP enrichment of TOP3Bccs following impairment of the ubiquitin-proteasomal pathway (MG132 or TAK243 treatment)[4]. To confirm the accumulation of TOP3Bcc in *TDRD3KO* cells, we performed IP experiments using TOP3B antibody with RADAR assay samples from MG132-treated wild type and *TDRD3KO* HCT116 cells. Higher endogenous TOP3Bccs could be detected in *TDRD3KO* HCT116 cells (Supplementary Fig. 4c).

To determine whether TDRD3 suppresses TOP3Bccs forming both in DNA and RNA, we isolated nucleic acids containing TOP3Bccs by RADAR assay[4] in cells with and without TDRD3. Next, we enriched TOP3Bccs by IP with HA antibody and digested the immunoprecipitated TOP3Bcc samples either with a mix of RNase A and RNase T1 to remove RNA, or with DNase 1 to digest DNA[4]. Slot-blot results showed that lack of TDRD3 enhances cellular TOP3Bccs both in DNA and RNA (Fig. 5d, e).

To address which domain of TDRD3 regulate TOP3Bcc levels, we co-transfected *TDRD3KO* HCT116 cells with HA-TOP3B and/or five different TDRD3 variants (Supplementary Fig. 4d, e): full-length TDRD3 (744 aa), TDRD3 transcript isoform 2 with a 93 amino acid N-terminal (651 aa) deletion, 1-352 TDRD3-FLAG (with a partially deleted linker domain and deleted Tudor domain, FMRP interaction domain and

exon junction binding motif) and 1-588 TDRD3-FLAG (with deleted Tudor domain FMRP interaction domain and exon junction binding motif) and 1-171 TDRD3-MBP and RADAR assays were performed to detect TOP3Bccs. Transfection with 1-352 TDRD3-FLAG and 1-588 TDRD3-FLAG mutants (having deletions in Tudor domain) lowered TOP3Bccs more or less with similar efficiency as full-length TDRD3 (Supplementary Fig. 4f, g). In contrast, TDRD3 transcript isoform 2 having partial deletions in N-terminal DUF-OB fold (first 93 amino acids; 651 aa) and 1-171 TDRD3-MBP having deletions in UBA domain could only partially reduce TOP3Bccs (Supplementary Fig. 4f, g). These results suggest that the N-terminal domain of TDRD3 (containing the DUF-OB fold, UBA, and the linker domain) is involved in limiting the accumulation of cellular TOP3Bccs.

As accumulation of abortive topoisomerase cleavage complexes is cytotoxic and the anticancer mechanism of action of topoisomerase I and II inhibitors[34,35,37], and abortive TOP3Bccs perturb different cellular processes and genomic stability[4], we studied R-loop formation, genomic damage and impediment of cell growth in the absence of TDRD3. To measure R-loops, we isolated genomic DNA from *TDRD3KO* HCT116 cells transfected with TOP3B alone or in combination with TDRD3 and performed Dot Blot assays with the S9.6 antibody[4,12,38]. In the absence of TDRD3, elevated TOP3Bccs were associated with increased cellular R-loops (Fig. 5f, g).

To examine genomic DNA damage, we measured histone H2AX phosphorylation (γH2AX)[39]. Increased TOP3Bccs in TDRD3-depleted cells was associated with γH2AX induction (Fig. 5h). Additionally, in the absence of TDRD3, TOP3B was found in association with PARP1 and other single-strand break repair (SSBR) and base excision repair (BER) proteins including XRCC1 and LIG3 (Supplementary Fig. 5a; and IP-MS experiment reported previously[12]). Because PARP1 recruitment to DNA lesions, auto-PARylation, and widespread PARylation are among the earliest responses to DNA damage[2,40–42], we measured PARylation in *TDRD3KO* HCT116 cells transfected with TOP3B construct alone or co-transfected with TDRD3. In the absence of TDRD3, TOP3B-transfected cells displayed increased global cellular PARylation both in the absence and presence of poly(ADP-ribose) glycohydrolase inhibitor (PARGi, PDD00017273, which prevents cellular dePARylation) in comparison to TDRD3-complemented cells (Supplementary Fig. 5b). To validate these results, we confirmed that this PARylation was suppressed by the PARP inhibitor talazoparib (Supplementary Fig. 5b). Collectively, these results demonstrate TOP3B-mediated DNA damage in *TDRD3KO* HCT116 cells.

To assess the global effect of the excessive TOP3Bccs observed in cells lacking TDRD3, we performed colony formation assays. TOP3B-transfected *TDRD3KO* cells exhibited reduced growth and defective colony formation phenotype compared to TDRD3-TOP3B co-transfected cells (Fig. 5i, j). From all these results we conclude that TOP3B in the absence of TDRD3 is deleterious and that MIB1-mediated ubiquitylation and degradation of TOP3B protects cells from the deleterious consequences of abortive TOP3Bccs.

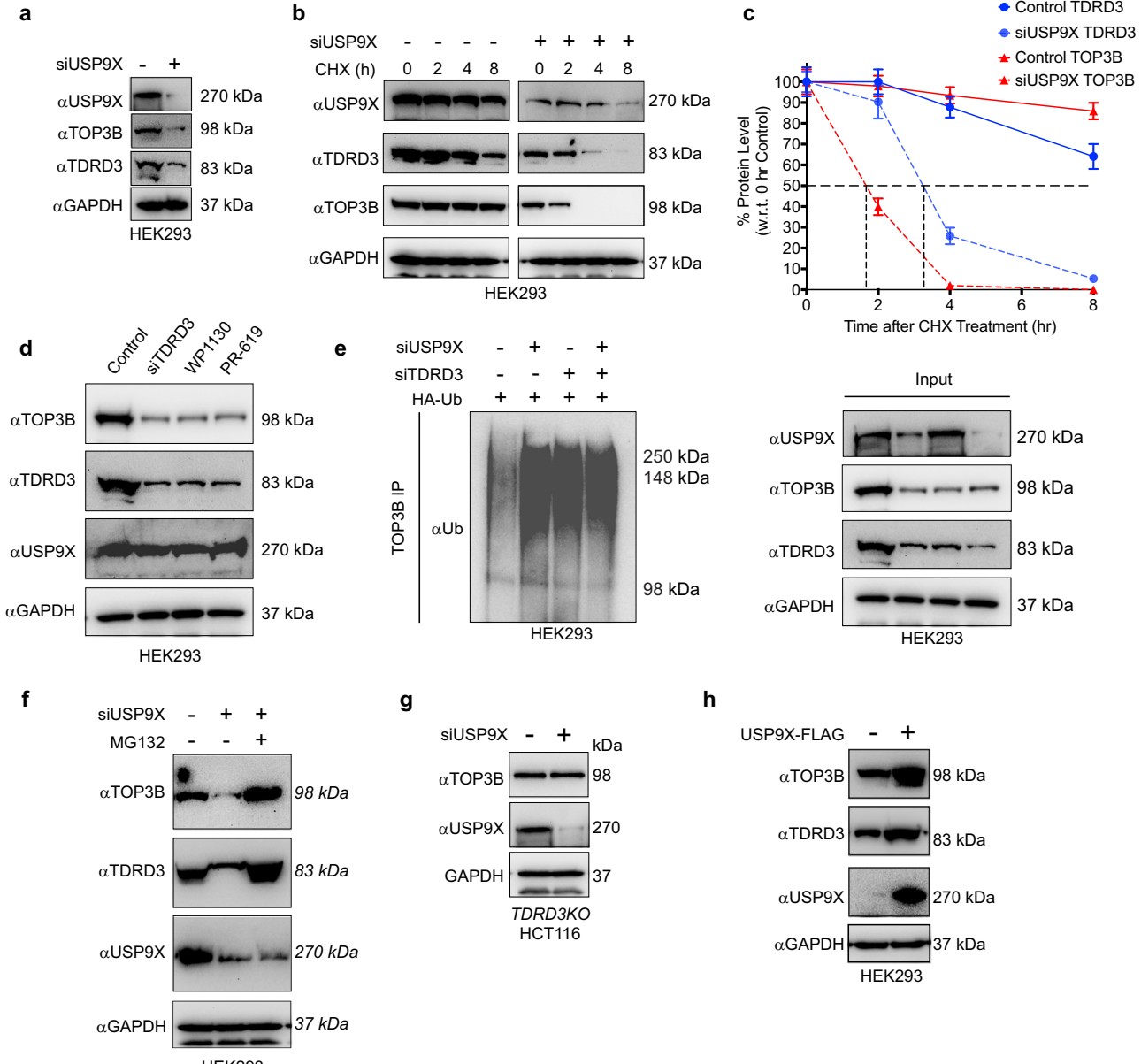

**Fig. 3 | USP9X deubiquitylates and protects TOP3B from proteasomal degradation. a** Loss of USP9X destabilizes both TOP3B and TDRD3 proteins in HEK293 cells. **b, c** Depletion of USP9X decreases the half-lives of both TDRD3 and TOP3B in HEK293 cells. Cells were transfected with USP9X siRNA and treated with CHX (10 μg/ mL) for the indicated time. Panel **b** is a representative Western blot and panel **c** the quantitation of TOP3B and TDRD3 protein levels from three independent experiments. Data are plotted as means ± standard deviations (SD). **d** Inhibition of USP9X with the DUB inhibitor Degrasyn (WP1130) decreases both TOP3B and TDRD3 protein levels in HEK293 cells. Cells were treated with Degrasyn (WP1130; 2 μM, 24 h), pan-DUB inhibitor PR619 (5 μM, 24 h) or transiently transfected with TDRD3 siRNA. **e** USP9X can deubiquitylate TOP3B and TDRD3-USP9X axis is important for the deubiquitylation process. HEK293 cells were transfected with HA-Ubiquitin construct and siTDRD3, siUSP9X or both (as indicated) for 48 h and subjected to TOP3B pulldown. Pulled down and input samples were resolved on SDS-PAGE and probed with ubiquitin, TOP3B, TDRD3, and USP9X antibodies (GAPDH as loading control for input samples). The figure is representative of three independent experiments. **f** Treatment with the proteasome inhibitor MG132 rescues the TOP3B protein levels in USP9X-depleted cells. HEK293 cells were transfected with USP9X siRNA for 48 h. Before harvest, cells were treated with MG132 (1 μM, 24 h) as indicated and lysates were subjected to western blotting with USP9X, TOP3B, and TDRD3 antibodies (GAPDH loading control). **g** USP9X downregulation does not affect TOP3B protein level in *TDRD3KO* HCT116 cells. After transfecting the cells with USP9X siRNA for 48 h, lysates were subjected to Western blotting with USP9X and TOP3B antibodies (GAPDH as loading control). **h** Ectopic expression of USP9X increases both TOP3B and TDRD3 protein levels. After transfection of HEK293 cells with USP9X-FLAG construct for 48 h, cell lysates were subjected to western blotting with TOP3B, TDRD3, and USP9X antibodies (GAPDH as loading control). Source data are provided as a Source Data file.

To check whether TOP3Bcc accumulation in cells lacking TDRD3 was associated with an upregulation of the TOP3Bcc excision phosphodiesterase TDP2[4], we checked TDP2 protein expression. We did not observe such upregulation after down-regulation of TDRD3 or in the absence of TDRD3 (Supplementary Fig. 6a, b). This result is consistent with our IP-Mass spec experiments showing no enrichment for TDP2 in the TOP3B interactome in the absence of TDRD3 protein.

Because of present and prior studies show that two E3 ligases, MIB1 and TRIM41[4] ubiquitylate TOP3B, we tested the individual roles of MIB1 and TRIM41. To address this point, we pulled down free TOP3B (no nuclease/benzonase in IP buffer) and total cellular TOP3B (benzonase in IP buffer) from HEK293 cells and looked for TRIM41 and MIB1 interaction. MIB1 interacted with both free and total cellular TOP3B whereas TRIM41 interacted only with total cellular fraction of TOP3B

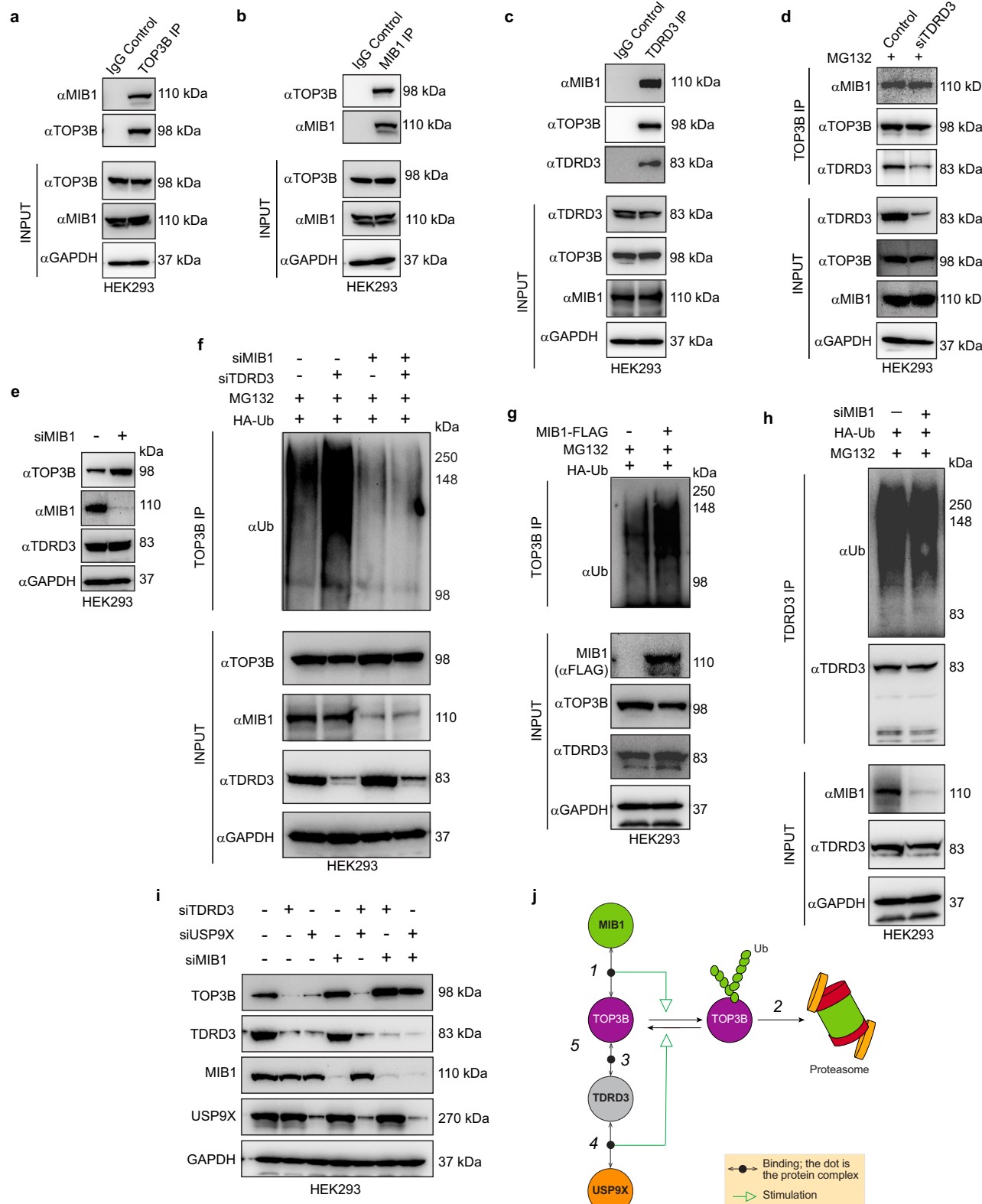

(new Supplementary Fig. 7a, b). This indicates that TRIM41 interacts with TOP3B in the presence of nucleic acids. To further this conclusion, we isolated chromatin fractions from HEK293 cells and pulled down TOP3B. Only TRIM41 could interact with chromatin-bound TOP3B (Supplementary Fig. 7c).

To find out the specific roles TRIM41 and MIB1 play in controlling TOP3Bcc levels in cells after TDRD3 depletion, we checked TOP3Bccs in FLAG-TOP3B-transfected *TDRD3KO* HCT116 cells after siRNA-mediated

down-regulation of TRIM41 and MIB1 in the absence of TDRD3. Down-regulation of both TRIM41 and MIB1 increased in TOP3Bcc levels (Supplementary Fig. 7d). To test whether MIB1 can ubiquitylate TOP3Bccs, we pulled down TOP3Bccs from RADAR assay samples prepared from FLAG-TOP3B-transfected *TDRD3KO* HCT116 cells after siRNA-mediated down-regulation of TRIM41 and MIB1. Only TRIM41 down-regulation affected TOP3Bcc ubiquitylation (Supplementary Fig. 7e). These results indicated that TRIM41 ubiquitylates TOP3Bccs and MIB1 free TOP3B.

**Fig. 4 | MIB1 ubiquitylates and drives the proteasomal degradation of TOP3B independently of TDRD3 and TDRD3-USP9X complex works downstream of MIB1 to limit TOP3B degradation. a** TOP3B pulldown-Western blot experiments showing TOP3B-MIB1 interaction in HEK293 cells. **b** MIB1 pulldown-Western blot experiments confirming TOP3B-MIB1 interaction in HEK293 cells. **c** TDRD3 pulldown-Western blot experiments showing TDRD3-MIB1 interaction in HEK293 cells. **d** TOP3B pulldown-Western blot experiments showing that TOP3B interacts with MIB1 both in wild type and siTDRD3-transfected HEK293 cells. **e** MIB1 depletion increases endogenous TOP3B protein level without affecting TDRD3 levels. **f** MIB1 ubiquitylates TOP3B independently of TDRD3. HEK293 cells were transfected with HA-Ubiquitin construct and siTDRD3, siMIB1 or both. Before harvest, cells were treated with MG132 (1 μM, 24 h) and subjected to TOP3B pulldown. Pulled down and input samples were resolved on SDS-PAGE and probed with Ubiquitin, TOP3B, TDRD3, and MIB1 antibodies (GAPDH as loading control). The figure is representative of three independent experiments. **g** Ectopic expression of MIB1 increases TOP3B ubiquitylation. HEK293 cells were transfected with HA-Ubiquitin construct and MIB1-FLAG construct. Before harvest, cells were treated with MG132

(1 μM, 24 h) as indicated and subjected to TOP3B pulldown. Pulled down and input samples were resolved on SDS-PAGE and probed with Ubiquitin, TOP3B, TDRD3, and MIB1 antibodies (GAPDH as loading control). The figure is representative of three independent experiments. **h** MIB1 depletion does not affect TDRD3 ubiquitylation. Cells were transfected with HA-Ubiquitin construct and siMIB1 as indicated for 48 h. Before harvest, cells were treated with MG132 (1 μM, 24 h) as indicated and subjected to TDRD3 pulldown. Pulled down and input samples were resolved on SDS-PAGE and probed with Ubiquitin, TDRD3 and MIB1 antibodies (GAPDH as loading control for input samples). The figure is representative of three independent experiments. **i** USP9X antagonizes MIB1 to stabilize TOP3B protein. HEK293 cells were transfected with the indicated siRNAs for 48 h. **j** Schematic representation of the homeostatic regulation of TOP3B protein levels by TDRD3-USP9X and MIB1. The TOP3B interacting E3 ligase MIB1 ubiquitylates and targets TOP3B for proteasomal degradation while the TDRD3-USP9X complex deubiquitylates and protects TOP3B from proteasomal degradation. Source data are provided as a Source Data file.

## TDRD3 stimulates the turnover of TOP3Bccs

To explain the increased accumulation of abortive TOP3Bccs in cells lacking TDRD3, we performed TOP3Bcc assays[4] with DNA or RNA oligonucleotide substrates labeled at their 3′-end with a fluorophore and with recombinant TOP3B (Fig. 6a). TOP3Bccs form covalent linkage with the 5′-end of cleaved DNA or RNA (TOP3Bccs) resulting in slow migrating bands in SDS-PAGE[1,4]. Measuring the steady-state levels of DNA and RNA TOP3Bccs in the absence and presence of TDRD3 showed that TDRD3 decreased the steady-state levels of DNA and RNA TOP3Bccs (Fig. 6a, b), consistent with the reported faster turn-over of the TOP3Bccs in the presence of TDRD3[24].

To demonstrate that TDRD3 acts by facilitating the rejoining of TOP3Bccs, we performed reversal kinetics experiments in the presence and absence of TDRD3. Steady-state DNA and RNA TOP3Bccs were prepared as described above in the absence and presence of TDRD3 (TOP3B: TDRD3 = 1: 4); after which, the samples were diluted 3-fold in reaction buffer to initiate the dissociation of TOP3B (Fig. 6c). In the absence of TDRD3, the DNA and RNA TOP3Bccs did not reverse significantly within 30 min. By contrast, in the presence of TDRD3, the DNA and RNA TOP3Bcc reversed (Fig. 6d). Together these results indicate that in the absence of TDRD3, DNA and RNA TOP3Bccs display slower reversal rate, which leads to the accumulation of the DNA and RNA TOP3Bccs and slower turnover rate for TOP3B. This result is consistent with earlier studies demonstrating stimulation of catalytic activity of TOP3B in presence of TDRD3[24,25].

## Discussion

The mechanistic details of TDRD3-mediated regulation of TOP3B have only been partially explored[6,7,10,24,25]. Our study addresses four main outstanding questions: 1/ How TDRD3 stabilizes TOP3B; 2/ What is the ubiquitin ligase for TOP3B; 3/ how TDRD3 affects the stability and turnover rate of TOP3Bccs in cells; and 4/ Why do cells target TOP3B for proteasomal degradation in the absence of TDRD3. This led us to identify a conserved MIB1-TDRD3-USP9X axis working in an orchestrated manner to maintain stoichiometric levels of TOP3B and TDRD3 in such way that free TOP3B tends to be removed and TOP3B-TDRD3 complexes are prominent in the cell. We show that the E3 ubiquitin ligase MIB1 interacts with and ubiquitylates TOP3B, leading to TOP3B proteasomal targeting and degradation. We also show that TDRD3 recruits the DUB USP9X to antagonize MIB1-mediated ubiquitylation. Finally, we propose that cells degrade TOP3B in the absence of TDRD3 to avoid the formation of unregulated DNA/RNA TOP3Bccs and subsequent genomic damage. Our model for the TOP3B homeostasis is summarized in Fig. 7.

We identified USP9X (Ubiquitin Specific Protease 9X) as the TDRD3-associated DUB for TOP3B by IP-MS experiments. USP9X is a member of the USP cysteine protease family of DUBs characterized by a conserved USP catalytic domain (CD-fold) resembling an overall

hand-like structure with fingers, thumb, and palm subdomains[28,43]. USP9X removes K11-, K63-, K48-, and K6-linked poly-ubiquitin chains from target proteins[43] controlling critical biological processes such as cell polarity, cell death/ apoptosis, neural cell development, embryogenesis and stem cell self-renewal[28,43–45]. USP9X is upregulated in different cancers (viz. myeloma, cervical, colorectal cancers)[28,43] and much like TOP3B and FMRP, depletion of USP9X is linked with neurological/ neurodevelopmental disorders and X-linked intellectual disability[44,45]. Although it has been reported that USP9X interacts with, deubiquitylates, and stabilizes TDRD3 and that USP9X is stimulated by TDRD3[27], our study reveals that TOP3B is a substrate of USP9X and establishes that USP9X requires TDRD3 to deubiquitylate TOP3B. Yet, USP9X may not be the only DUB for TOP3B, as our TOP3B IP-MS experiments identified other deubiquitylases (DUBs) including USP39, USP7, USP10 in addition to USP9X. Further studies are therefore warranted to determine whether other DUBs can deubiquitylate TOP3B under different cellular scenarios and whether they require TDRD3 to interact with and work on TOP3B similar to USP9X.

It is also not excluded that TDRD3 protects TOP3B from proteasomal degradation by additional mechanisms. TDRD3 contains an internal ubiquitin-associated (UBA) polypeptide domain (amino acid residues 286-328)[46] between its DUF-OB fold and Tudor domains, and it is still not known whether this UBA domain can interact with ubiquitylated TOP3B. Apart from interacting with Lys48-linked tetra-ubiquitin chains, the function of the UBA domain of TDRD3 is not well defined[20,22,23].

Our results also reveal that MIB1 (Mind bomb 1) interacts with free TOP3B and acts as a prominent E3 ubiquitin ligase for TOP3B leading to the proteasomal degradation of TOP3B. First, knocking down *MIB1* reduced the ubiquitylation of TOP3B and increased the stability of cellular TOP3B. Second, overexpression of MIB1 increased the cellular ubiquitylation of TOP3B while reducing TOP3B levels. MIB1 is a prominent member of the RING domain containing family of E3 ubiquitin ligases containing three carboxy-terminal RING finger domains[47,48]. It was first identified during zebrafish mutagenesis screens as MIB1 mutants displayed defective brain morphology and neurogenesis (because of impaired Notch activation)[47–50]. MIB1 is a bona fide E3 ubiquitin ligase mediating ubiquitylation and degradation of different protein substrates including SMN, WRN, and receptor-like tyrosine kinase (RYK)[51–53].

Previously we found another E3 ligase, TRIM41, which unlike MIB1 (which can work on free TOP3B) ubiquitylates trapped TOP3Bccs on DNA and RNA[4]. It is intriguing that different E3 ligases ubiquitylate TOP3B in different conditions. Our study demonstrates the differential activities of the two TOP3B E3 ligases, with TRIM41 ubiquitylating TOP3Bccs to allow TDP2 activity[12] while MIB1 ubiquitylates free TOP3B that tends to form TOP3Bccs (Fig. 7).

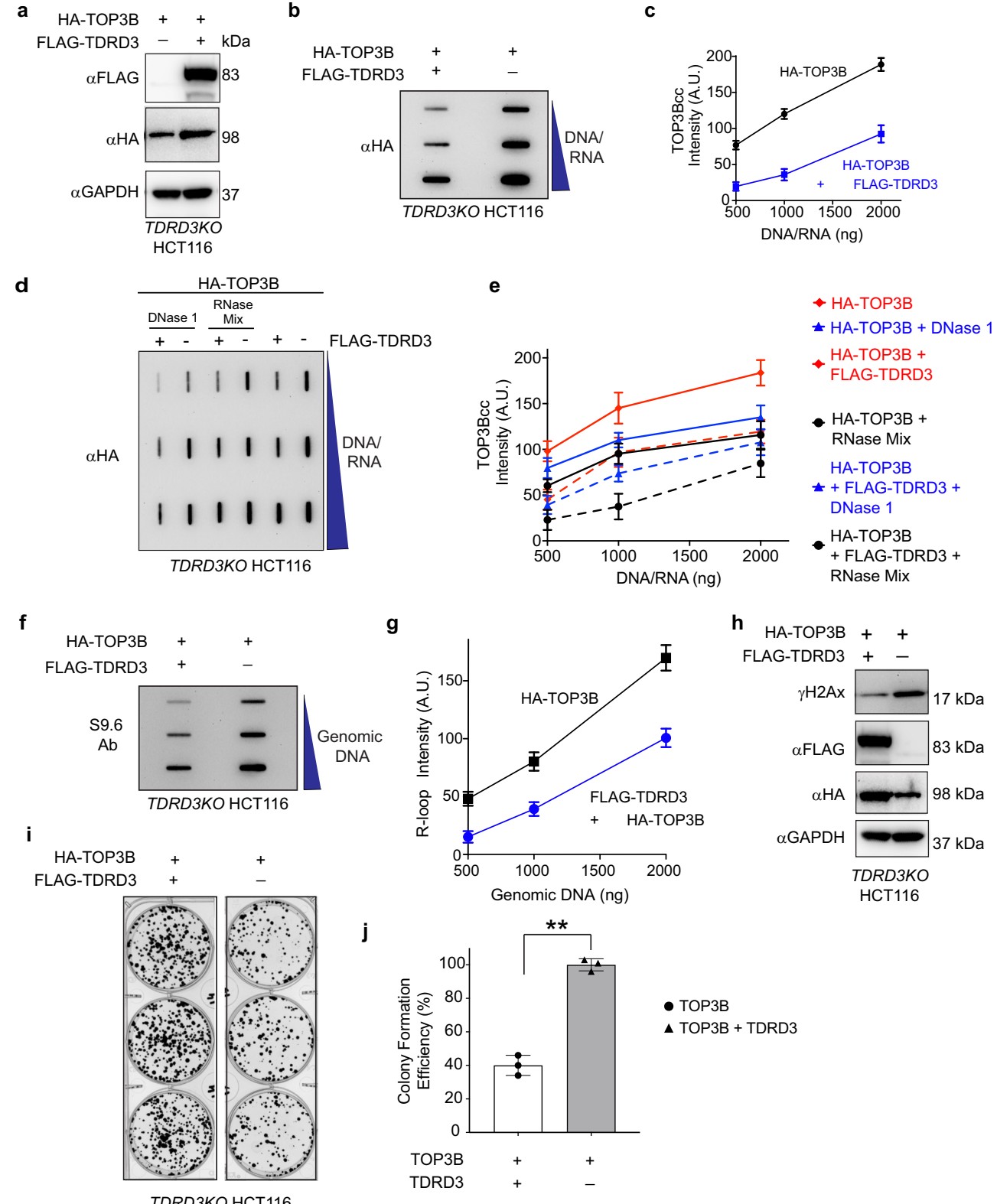

Topoisomerases work in cells by breaking and resealing nucleic acids with the formation of a transient enzyme-nucleic acid complex via a phosphodiester bond between their catalytic tyrosine residues and one of the ends of the broken nucleic acid (3′-end for type IB and 5′-end for type IA and type II topoisomerases). Under normal conditions, these enzyme-nucleic acid intermediates, which are referred to as "Topoisomerase cleavage complexes" (TOPccs) readily reverse, which allows the resealing of the nucleic acid backbone after

topoisomerase activity[1,3]. When these TOPccs fail to reverse, they lead to topoisomerase-linked single- or double-strand DNA breaks, which are cytotoxic and genotoxic, and are the target of some of the most potent anticancer drugs targeting TOP1ccs and TOP2ccs[34,35,37]. Our cellular and biochemical experiments suggest that, in the absence of TDRD3, TOP3Bccs tend to be trapped on DNA and RNA. Therefore, we propose that TDRD3 plays a critical role in preventing the accumulation of abortive TOP3Bccs[4,12], which otherwise induce R-loops,

**Fig. 5 | Cellular depletion of TDRD3 causes accumulation of aberrant TOP3Bccs and genomic instability. a** Representative immunoblots showing expression of HA-TOP3B and FLAG-TDRD3 in *TDRD3KO* HCT116 cells transfected for 48 h. **b, c** Increased TOP3Bccs in the absence of TDRD3. RADAR assay samples were prepared from *TDRD3KO* HCT116 cells transfected with HA-TOP3B alone or together with TDRD3-FLAG constructs (as indicated). Samples were immunoprecipitated with anti-HA antibody, eluted from the beads, ethanol precipitated and resuspended. Samples were slot blotted and TOP3Bccs were detected with anti-HA antibody. Panel **b** displays a representative slot blot. Panel **c** is the quantitation of TOP3Bcc formation from three independent experiments. Data are means ± SD. A.U. is arbitrary units. **d, e** TDRD3 depletion enhances both DNA and RNA TOP3Bccs. Protein-nucleic acid adducts were isolated by RADAR assay from *TDRD3KO* HCT116 cells transfected with HA-TOP3B alone or together with TDRD3-FLAG constructs (as indicated). TOP3Bccs were enriched by IP using HA antibody. Samples were digested either with excess RNase A (200 µg/mL) and RNase T1 (200 units/ml) mix, or with DNase 1 (10 units), ethanol-precipitated, resuspended and slot-blotted. TOP3Bccs were detected with anti-HA antibody. Representative slot blot of DNA and RNA TOP3Bccs is displayed in panel d. Panel e is the quantitation of TOP3Bccs from independent experiments. Data are means ± standard deviations (SD; *n* = 3). A.U. is arbitrary units. **f, g** R-loop accumulation in TDRD3-depleted cells. transfected with TOP3B Genomic DNA isolated from *TDRD3KO* HCT116 cells transfected as indicated was slot blotted and probed with S9.6 antibody. Panel G is the quantitation from three independent experiments. Data are means ± SD. A.U. is arbitrary units. **h** γH2AX induction in TDRD3-depleted cells transfected with TOP3B. Immunoblots showing γH2AX levels in *TDRD3KO* HCT116 cells transfected with HA-TOP3B alone or together with TDRD3-FLAG constructs (as indicated). **i, j** Representative images of colony formation assay of *TDRD3KO* HCT116 cells transfected with HA-TOP3B alone or together with TDRD3-FLAG constructs (panel **i**). Quantitative representation of colony formation assays as shown in left panel (panel **j**). Data are provided as means ± SD (*n* = 3). Two-tailed unpaired *t*-test with Welch's correction. **P value = 0.0004. Source data are provided as a Source Data file.

genomic breaks (γH2AX activation) and growth defect. This could explain the importance of TDRD3 in controlling TOP3B activity, and why cells have evolved ways for degrading TOP3B in the absence of TDRD3.

Loss of TDRD3 also led to a detectable interaction of TOP3B with PARP1, XRCC1, and LIG3 and this increased interaction of TOP3B with PARP1, XRCC1 and LIG3 in the absence of TDRD3 is possibly caused by the increased demand to repair TOP3Bccs and the resulting double-strand breaks (DSBs). It will be interesting to map TOP3Bcc sites in cells in presence and absence of TDRD3. As TOP3B can be recruited to and resolve R-loops[10,12] and R-loops are more prone to form at promoters or transcription termination regions, it is plausible that in absence of TDRD3, TOP3Bccs and the resulting DSBs can form in these genomic regions.

## Methods
### Cell Lines
HEK293 (Cat# CRL-1573, ATCC, Manassas, VA) and HCT116 (Developmental Therapeutics Program, National Cancer Institute) cell lines were grown in Dulbecco's modified Eagle's medium (DMEM; ThermoFisher Scientific, Cat# 11965-092) supplemented with 10% Fetal Bovine Serum (Gemini, West Sacramento, CA, Cat# 100-106) and 1% penicillin-streptomycin (ThermoFisher Scientific, Cat# 15140-122) at 37 °C in humidified 5% CO2 chamber. During subculture cells were detached using Trypsin-EDTA (0.05%, ThermoFisher Scientific, Cat# 25300054). Tni-FNL cells were cultured in Gibco Express 5 medium with 18 mM glucose. Generation of *TDRD3KO* HCT116 cells were reported previously[5]. Cell lines were tested for mycoplasmas (MycoAlert kit – 100 tests, Lonza, LT07-318). Cell line authentication was carried out using short tandem repeat analysis at Frederick National Laboratory, NCI-NIH.

### Mammalian expression constructs and transient expression in mammalian cell
Human TOP3B-Myc-FLAG cDNA ORF (CAT#: RC223204) Clone were purchased from OriGene. The full-length cDNAs of TOP3B was PCR-amplified from TOP3B-Myc-FLAG cDNA ORF Clone (CAT#: RC223204) respectively using cloning primers (TOP3B forward primer 5'-GCTTGGATCCCAAGACTGTGCTCATGGTT-3'; TOP3B reverse primer 3'-CCAAGAATTC TCATACAAAGTAGGCGGC-5') and subcloned into pcDNA3-HA with BamHI and EcoRI sites. Human TDRD3-Myc-FLAG cDNA ORF (CAT#: RC228569), TDRD3 (Myc-DDK-tagged)-Human tudor domain containing 3 (TDRD3), transcript variant 2 (CAT#: RC207081), Human MIB1-Myc-FLAG cDNA ORF (CAT#: RC221377) and Human USP9X-Myc-FLAG cDNA ORF (CAT#: RC217531) Clones were purchased from OriGene. HA-Ubiquitin WT plasmid was a gift from Ted Dawson (Addgene plasmid CAT#: 17608). Plasmids were transfected in HCT116 and HEK293 cells using Lipofectamine 3000 Reagent (CAT#: L3000015, ThermoFisher Scientific) according to the manufacturer's protocol for 48–72 h.

### Site-Directed Mutagenesis and generation of deletion mutants in Mammalian Expression Vectors
Site-Directed Mutagenesis (SDM) was performed using QuikChange II XL site-directed mutagenesis kit (Agilent Technologies) following the manufacturer's protocol.

C1566S USP9X active site mutant was generated using oligonucleotides: USP9X_FP: 5'-GAATCACAGAATTCATGTAACTAGTAGCACC GGCATTTTTCAG-3' and USP9X_RP: 5'-CTGAAAAATGCCGGTGCTACT AGTTACATGAATTCTGTGATTC-3'. C985S MIB1 active site mutant was generated using oligonucleotides: MIB1_FP: 5'-TCATGCGGTCTCC ACTGAGTTGACAGGTTCC-3' and MIB1_RP: 5'-GGAACCTGTCAACTC AGTGGAGACCGCATGA-3' (Supplementary Table 2).

Ubiquitin K6R was generated by Q5 SDM Kit using oligonucleotide 5'-ATCTTCGTGAGGACCCTGACTGG-3'. Ubiquitin K11R was generated using oligonucleotide 5'-CTGACTGGTAGGACCATCACTC-3'. Ubiquitin K27R was generated using oligonucleotide 5'-GAGAATGTCAGGG-CAAAGATCC-3'. Ubiquitin K29R was generated using oligonucleotide 5' GTCAAGGCAAGGATCCAAGAC-3'. Ubiquitin K33R was generated using oligonucleotide 5'-ATCCAAGACAGGGAAGGCATC-3'. Ubiquitin K48R was generated using oligonucleotide 5'-TTTGCTGGGAGA-CAGCTGGAA-3'. Ubiquitin K63R was generated using oligonucleotide 5'-AACATCCAGAGAGAGTCCACCC-3'.

C-terminal deletion mutants of human TDRD3 i.e., 1-352 TDRD3 and 1-588 TDRD3 were generated using Human TDRD3-Myc-FLAG (OriGene; CAT#: RC228569) as PCR template and oligonucleotides: TDRD3_F- 5'-CGCCATGGCCCAGGTGGCCGGCGCG-3', TDRD3(352) _R- 5'-CGCGTAGATCTTATTCGCCCCCTGCC-3' and TDRD3(588) _R- 5'-CGCGTACCAATGAAACTATTACTTCG-3' (Supplementary Table 2).

cDNA for human TDRD3 (1-171) was PCR amplified with primers 5'-CAGAATTCGCCACCATGGCCCAGGTGGCCGGCGCGG-3' and 5'-CAGCGGCCGCATTGTGTTTTGATAAGCTTCTC-3' (Supplementary Table 2), inserted into a modified pLEXm plasmid, including a C-terminal MBP-tag, with NotI and EcoRI sites.

### Recombinant human TOP3B production
Recombinant TOP3B was purified from baculovirus system[4,12]. Briefly, TOP3B was initially PCR amplified from Human TOP3B-Myc-FLAG cDNA ORF (CAT#: RC223204) using forward primer: 5'-CGGGGTAC-CATGAAGACTGTGCTCATGG-3' and reverse primer: 5'-CCGCTCGAG TCATACAAAGTAGGCGGCCAG-3' (Supplementary Table 2) and cloned into Gateway entry vector pENTR3C (Invitrogen, CAT#: A10464). TOP3B was then subcloned by Gateway LR recombination (Thermo Fisher) into pDest-635 (22876-X01-635) for insect cell expression which includes an N-terminal His6X tag. Bacmid was prepared in DE77, a DH10Bac-derived strain (Bac-to-Bac system, Thermo Fisher) and after

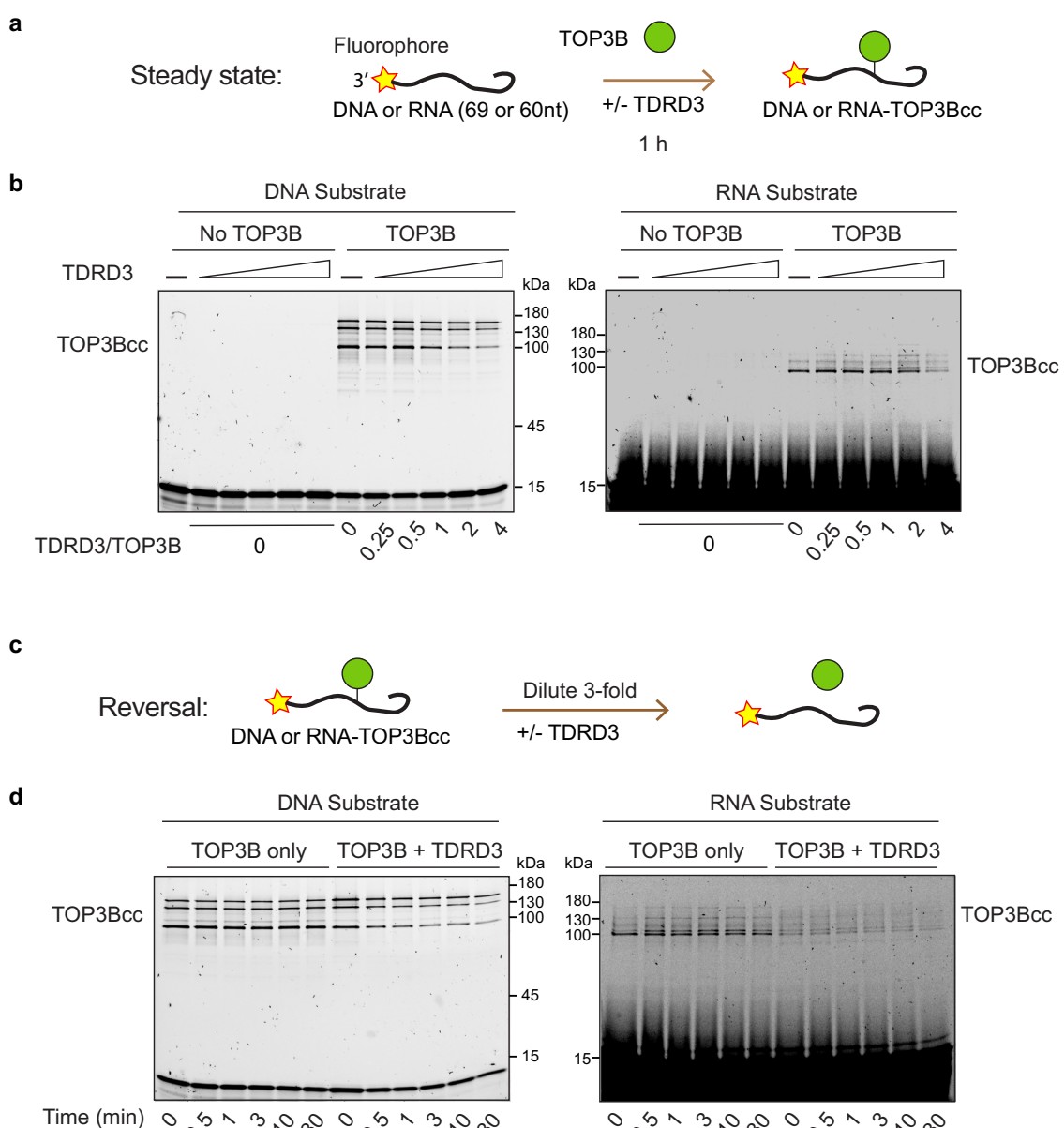

**Fig. 6 | TDRD3 prevents the accumulation of TOP3B DNA and RNA cleavage complexes. a** Scheme of the in vitro TOP3Bcc formation assay in steady-state condition. DNA or RNA oligo substrate labeled at the 3′-end with a fluorophore is incubated with recombinant TOP3B in the presence or absence of recombinant TDRD3. TOP3Bcc formation results in a slower migrating band and can be detected via the fluorophore at the 3′-ends of oligo construct[4]. **b** Reduced steady-state levels of DNA and RNA TOP3Bccs with increasing TDRD3 (stoichiometric ratio indicated below the gel). **c** Scheme of the TOP3Bcc reversal kinetics assay. TOP3Bccs formed (as indicated above) in the absence and presence of TDRD3 (TOP3B: TDRD3 = 1: 4) were diluted 3-fold and allowed to reverse for 0–30 min. **d** Time-course showing the stimulation of DNA and RNA TOP3Bcc reversal by TDRD3. Source data are provided as a Source Data file.

purification, bacmid DNA was verified by PCR amplification across the bacmid junctions. Bacmids were transfected in SF-9 cells using PEI (1 mg/ml with 5% glucose; Polysciences, CAT#: 23966), recombinant baculovirus stock was collected and titrated using ViroCyt (Beckamn). Two liters of Tni-FNL cells were set in a baffled 5-liter Thomson Optimum Growth Flask in Gibco Express 5 medium with 18 mM glucose at a cell density of 1 ×106 cells/ml at 27oC and 24 hrs later infected at a MOI (multiplicity of infection) of 3. After 3 days of incubation at 21oC, cell pellets were collected by centrifugation at 2000 rpm for 11 min and flash frozen on dry ice. Cell pellet was thawed by the addition of 200 ml of lysis buffer (20 mM HEPES, 300 mM NaCl, 1 mM TCEP and 1:100 v/v of Sigma protease inhibitor P8849) and homogenized by vortexing. The cells were lysed by performing two passes on an M-110EH-30 microfluidizer (Microfluidics) at 7000 psi, clarified at 100 K x g for

30 minutes at 4 °C using an optima L-90K ultracentrifuge (Beckman), filtered (0.45 micron) and applied to a f20 mL IMAC HP column (GE Scientific) that was pre-equilibrated with lysis buffer containing 50 mM imidazole on a Bio-Rad NGC. Column was washed with lysis buffer containing 50 mM imidazole and proteins were eluted with lysis buffer containing 500 mM imidazole. After SDS-PAGE/Coomassie staining, positive fractions were pooled, dialyzed to 20 mM HEPES, 50 mM NaCl, 1 mM TCEP, 0.5 mM PMSF, 1:1000 v/v of PI, 50% glycerol, pH 7.2. Protein concentration was determined (0.88 mg/ml) and stored at −80 °C for future use.

**Recombinant human TDRD3 production**
Recombinant TDRD3 was purified from baculovirus system. Briefly, tev-Hs.TDRD3 PCR product was generated using TDRD3-Myc-FLAG

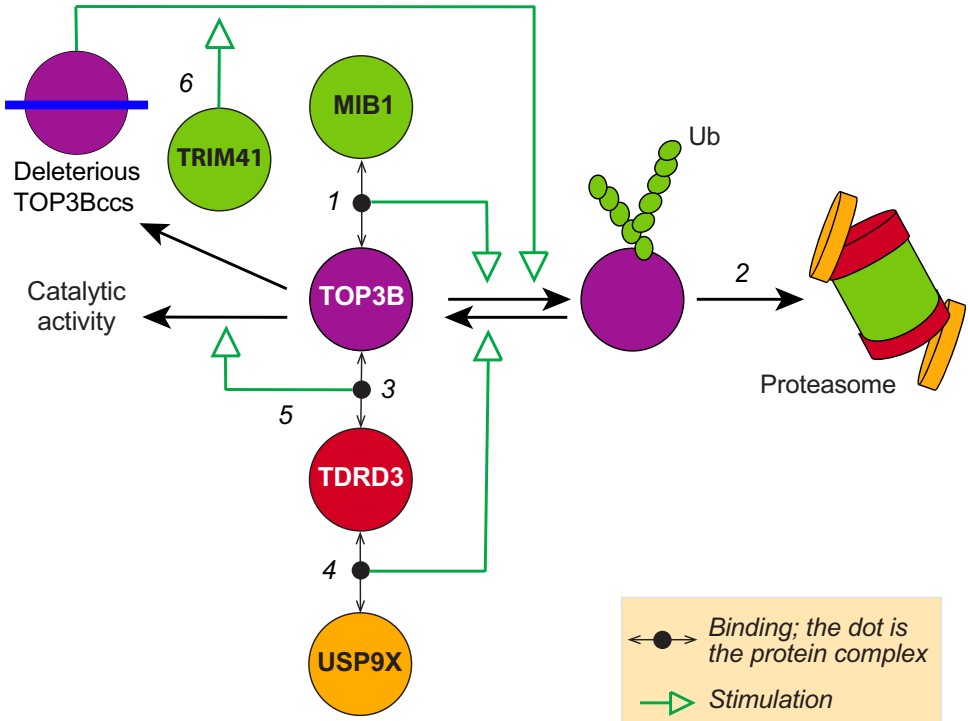

**Fig. 7 | Homeostatic regulation of TOP3B by TDRD3, USP9X, and MIB1.** (1) Free TOP3B is ubiquitylated by MIB1. (2) Ubiquitin (Ub)-modified TOP3B is degraded by the proteasome. (3) The binding of TOP3B to TDRD3 recruits the deubiquitinase USP9X (4), which stabilizes TOP3B by preventing its proteasomal degradation. (5) It also increases the turnover of TOP3B. (6) Excessive TOP3Bccs formed in the absence of TDRD3 are ubiquitylated by TRIM41 prior to their proteasomal processing and excision by TDP2 (not shown).

cDNA ORF (CAT#: RC228569) as template and the following primers: 23674TDRD3F (forward primer) 5′-GGGGACAACTTTGTACAAAAA AGTTGGCGAAAACCTGTACTTCCAAGGCATGGCCCAGGTGGCCGGCG C-3′ and 23675TDRD3R (reverse primer) 5′-GGGGACAACTTTGTAC AAGAAAGTTGATTAGTTCCGAGCCCGGGGTGGTT-3′ (Supplementary Table 2). PCR product was cloned into pDONR-255 via Gateway Recombination Cloning and the resulting Entry clone (29012-E02 tev-Hs.TDRD3) was then cloned into pDest-635 via Gateway Recombinational Cloning to obtain the final Expression clone (29012-X02-635 His6-tev-Hs.TDRD3). Expression clone was transformed into DE95 Bacmid cell line and bacmid clones were selected and grown in small culture. DNA was purified via alkaline plasmid preparation and a correct clone was verified by in house Bac check PCR. *Spodoptera frugiperda* (Sf9) cells were transfected with bacmid DNA using Cellfectin (Thermo Fisher Scientific, Cat#: 10362100). After five days of growth at 27 °C, baculovirus was harvested and titered. *Trichoplusia ni* (TniFNL) cells were infected with baculovirus at an MOI of 3. The culture was grown in a 21 °C shaker for 72 hours before harvesting the cells. Cell pellet from 1 liter of TniFNL culture was resuspended in 100 ml of 20 mM HEPES, pH 7.4, 300 mM NaCl, 1 mM TCEP, 1:100 (v:v) Sigma Protease Inhibitor Cocktail and lysed by passing through a microfluidizer twice at 7,000 psi. The lysate was clarified by ultracentrifugation at 100,000 x g at 4 °C for 30 minutes. The cleared lysate was filtered through a 0.45 µm Whatman PES syringe filter, and had imidazole added to a final concentration of 50 mM. All purification steps were performed on a Bio-Rad NGC chromatography system at room temperature. A 20-ml HisTrap HP column (Cytiva) was equilibrated in 10 column volumes (CV) Buffer A (20 mM Hepes, pH 7.4, 300 mM NaCl, 1 mM TCEP, 50 mM imidazole). Cleared lysate sample was loaded onto the column followed by 10 CV of Buffer A to wash the column. Protein was eluted from the column with 20 CV gradient from Buffer A to Buffer B (20 mM Hepes, pH 7.4, 300 mM NaCl, 1 mM TCEP, 500 mM imidazole). Fractions were collected throughout the purification and analyzed by SDS-PAGE/Coomassie Blue staining. Elution fractions containing TDRD3 were pooled. Pool concentrated using 10 K MWCO spin concentrator (Amicon) in preparation for size exclusion chromatography. A 16/60 HiLoad S200 SEC column (Cytiva) was equilibrated in 1.25 CV Buffer C (20 mM HEPES, pH 7.4, 300 mM NaCl, 1 mM TCEP). The concentrated sample was injected on to the column and eluted with 1.25 CV Buffer C. Fractions were collected throughout the purification and analyzed by SDS-Page/Coomassie Blue staining. Fractions containing TDRD3 were pooled and dialyzed into Buffer D (20 mM Hepes, ph 7.4, 300 mM NaCl, 1 mM TCEP, 10% glycerol) using 10 kDa molecular weight cut off SnakeSkin (Thermo Scientific).

### siRNA transfection
Silencing of TDRD3, USP9X and MIB1 were done using ON-TARGETplus SMARTpool siRNA targeting TDRD3 (CAT#: L-014655-01-0010, Dharmacon), ON-TARGETplus SMARTpool siRNA targeting MIB1 (CAT#: L-014033-00-0005, Dharmacon), ON-TARGETplus SMARTpool siRNA targeting TRIM41 (CAT#: L-007105-00-0005) and ON-TARGETplus SMARTpool siRNA targeting USP9X (CAT#: L-006099-00-0010, Dharmacon) respectively. All siRNAs were used at a final concentration of 25 nM and transfected using Lipofectamine® RNAiMAX transfection reagent (CAT#: 13778150, ThermoFisher Scientific) following the manufacturer's protocol for 48–72 h.

### Western blotting and antibodies
To prepare whole cell lysates for western blotting, cells were resuspended with RIPA buffer (150 mM NaCl, 1% NP- 40, 0.5% Sodium deoxycholate, 0.1% SDS, 50 mM Tris pH 7.5, 1 mM DTT and cOmplete Mini, EDTA-free protease inhibitor cocktail (Roche, Cat# 11836170001). After thorough mixing, samples were agitated at 4 °C for 30 min, sonicated for 30 s with 40% pulse (3-5 times), centrifuged at 15,000 × *g* at 4 °C for 15 min, and supernatant was collected.

For detection of total cellular TOP3B (both free and nucleic acid bound), cells were washed with PBS and incubated on a shaker with IP lysis buffer (50 mM Tris-HCl pH 7.4, 150 mM NaCl, 1 mM EDTA, 1% NP-40, 0.2% Triton X-100, 5% glycerol, 1 mM DTT, 20 mM *N*-ethylmaleimide (Millipore Sigma, Cat# E3876-25G) and protease inhibitor cocktail supplemented with 2 μl benzonase (250 units/μl, Sigma-Aldrich, CAT#: E8263) followed by 30 min incubation in ice. After brief sonication (40% power for 30 s pulse and 1 min rest 4 times), samples were centrifuged at 15,000×*g* at 4 °C for 30 min and the supernatant was collected[4,12].

Lysed samples were mixed with tris-glycine SDS sample buffer (Novex, LC2676) and loaded onto Novex tris-glycine gels (Novex). Blotted membranes were blocked with 5% non-fat dry milk in PBS with 0.1% Tween-20 (PBST). The primary antibodies were diluted in 5% milk in PBST by 1:1000 for Mouse monoclonal anti-FLAG M2 (Millipore Sigma, St. Louis, MO, CAT#: F1804), 1:10000 for Rabbit monoclonal anti-GAPDH (Cell Signaling Technology, Danvers, MA, CAT#: 2118 S), 1:1000 for Rabbit monoclonal anti-HA (Cell Signaling Technology, Danvers, MA, CAT#: 3724 S), 1:1000 for Rabbit monoclonal anti-TDRD3 (Cell Signaling Technology, Danvers, MA, CAT#: 5942 S), 1:1000 for Mouse monoclonal anti-MIB1 antibody (B9, Santa Cruz Biotechnology, Santa Cruz, CA, CAT#: sc-393811), 1:1000 for Rabbit monoclonal anti-TOP3B antibody (abcam, Waltham, MA, CAT#: ab183520), 1:1000 for Rabbit monoclonal anti-USP9X antibody (Cell Signaling Technology, Danvers, MA, CAT#: 14898 S), 1:2000 for Mouse monoclonal anti-phospho (S139)-H2AX (JBW301, Millipore Sigma, St. Louis, MO, CAT#: 05-636), 1:1000 for Mouse monoclonal Anti-PAR Polymer Monoclonal Antibody (R&D Systems, Inc. a Bio-Techne Brand, CAT#: 4335-MC-100), 1:1000 for Ub (P4D1) antibody (Cat# sc-8017, Santa Cruz Biotechnology), 1:1000 for Rabbit polyclonal TRIM41 antibody, (Cat# ab111580, Abcam). Secondary antibodies (anti-mouse IgG ECL, HRP conjugated, Cat# NA9310 and ant−rabbit IgG ECL, HRP conjugated, Cat# NA9340) (Supplementary Table 1) were diluted (1:10000) in 5% non-fat milk in PBST and signal was detected by ECL chemiluminescence reaction (SuperSignal™ West Femto Maximum Sensitivity Substrate, Thermo Scientific, Waltham, MA, Cat# 34095).

**Cycloheximide chase experiment**
Wild-type and siTDRD3 or siUSP9X transfected HEK293 cells and wild-type and *TDRD3KO* HCT116 cells were treated with cycloheximide (CHX; 10 μg/ml) for 0, 2, 4, and 8 h before harvesting. Cells were lysed, run on SDS-PAGE, and analyzed by western blot experiments (as described above) for indicated proteins (TOP3B and TDRD3).

**qRT-PCR.** RNA was isolate using pure link RNA mini kit Invitrogen according to manufactures instruction. The isolated RNA was immediately used to set up RT- PCR to quantify the expression of TOP3B gene using following primer set: TOP3B forward primer- 5′-GATGCTG GAGAAGCAGACGAAC-3′, TOP3B reverse primer-5′-CTCTCCACCGT-GACATAGTTGC-3′, Actin forward primer- 5′-TGCTATGTTGCCCTA-GACTTCG-3′, Actin reverse primer- 5′-GTTGGCATAGAGGTCTTT AC GG-3′. In brief, Luna one-step qRT-PCR kit was used to perform the reaction. 20ul reaction was set upped containing 750 ng RNA template, 10 μl Luna Universal One-Step Reaction Mix (2X), 1 μl Luna Warm Start RT Enzyme Mix (20X), 0.4 μM of each primer and Nuclease-free Water. The reactions tube was incubated in Quant 5 studio (Applied Biosystems/Thermo Fisher Scientific) and SYBR scan setting mode was used. Reverse transcription was performed for 10 minutes at 55 °C. The samples underwent 40 cycles of denaturation (10 seconds at 95 °C) and annealing/extension (30 seconds at 60 °C). Each sample was run in triplicate.

**Immunoprecipitation**
Cells were washed with PBS and incubated on a shaker with IP lysis buffer (50 mM Tris-HCl pH 7.4, 150 mM NaCl, 1 mM EDTA, 1% NP-40,

0.2% Triton X-100, 5% glycerol, 1 mM DTT, 20 mM N-ethylmaleimide and protease inhibitor cocktail) supplemented with 2 μl benzonase (250 units/μl, Sigma-Aldrich, CAT#: E8263) followed by 30 minutes incubation in ice. After brief sonication (40% power for 30 s pulse and 1 min rest 3 times), samples were centrifuged at 15,000 X g at 4 °C for 30 min and the supernatant was collected and an aliquot (20 μl) of the supernatant was saved as input. The rest of the supernatant was diluted in IP lysis buffer containing desired antibody (3 μg/ tube) and rotated overnight at 4 °C. Next day, 30 μl Protein A/G magnetic beads (ThermoFisher Scientific Cat#: 26162) were added and incubated with the lysates for anther 4 hrs. After magnetic separation, beads were washed with RIPA buffer 2 times. Finally, beads were resuspended in Tris-Glycine SDS Sample Buffer (2X) (ThermoFisher Scientific Novex™; Cat#: LC2676) for SDS-PAGE and immunoblotted with different antibodies as indicated or further processed for MS analysis.

**RADAR Assay and detection of DNA and RNA TOP3Bccs**
RADAR assay was performed for detection of TOP3Bccs as described previously[4]. HA or FLAG-tagged WT TOP3B transfected or MG132/ NSC690634/ NSC96932 treated HCT116/ HEK293 cells (1 × 10⁶) treated were harvested after 48–72 h, washed with PBS, and lysed by adding 1 mL DNAzol (ThermoFisher Scientific, CAT#:10503027). Nucleic acids were precipitated following addition of 0.5 mL of 100% ethanol, incubation at −20 °C for 5 min and centrifugation (12,000 × g for 10 min). Precipitates were washed twice in 75% ethanol, resuspended in 200 μl TE buffer, heated at 65 °C for 15 min, followed by shearing with sonication (40% power for 15 s pulse and 30 s rest 5 times). Samples were centrifuged at 15,000 rpm for 5 min and the supernatant containing nucleic acids with covalently bound proteins were collected. Nucleic acid containing protein adducts were quantitated, slot-blotted and TOP3Bccs were detected with either rabbit monoclonal Anti-TOP3B antibody [EP7779]−C-terminal (Abcam, CAT#: ab183520) or mouse monoclonal anti-FLAG M2 antibody (Millipore Sigma, St. Louis, MO, CAT#: F1804) or rabbit Anti-HA-Tag Monoclonal Antibody, Unconjugated, Clone C29F4 (Cell Signaling Technology, CAT#: 3724).

For immunoprecipitation and enrichment of RADAR assay samples, RADAR samples were diluted in IP buffer (50 mM Tris-HCl pH 7.4, 150 mM NaCl, 1 mM EDTA, 1% NP-40, 0.2% Triton X-100, 5% glycerol, 1 mM DTT, 20 mM N-ethylmaleimide and protease inhibitor cocktail) containing anti-TOP3B antibody (abcam, Cat# ab183520) and rotated overnight at 4 °C. Next day, Pierce™ ChIP-grade Protein A/G Magnetic Beads was added and incubated with the samples for another 4 hrs. Immunoprecipitated samples were washed with IP buffer twice, resuspended in TE buffer containing 1% SDS. Nucleic acids containing TOP3Bccs were precipitated following addition of 100% ethanol and centrifugation (12,000 × g for 10 min). Precipitates were washed twice in 75% ethanol, resuspended in TE buffer. Equal volumes of resuspended nucleic acid containing protein adducts were slot-blotted and TOP3Bccs were detected with with either rabbit monoclonal Anti-TOP3B antibody [EP7779]−C-terminal (Abcam, CAT#: ab183520) or mouse monoclonal anti-FLAG M2 antibody (Millipore Sigma, St. Louis, MO, CAT#: F1804) or rabbit Anti-HA-Tag Monoclonal Antibody, Unconjugated, Clone C29F4 (Cell Signaling Technology, CAT#: 3724).

For detection of DNA and RNA TOP3Bccs, 10 μg RADAR assay samples were digested either with excess RNase A (200 μg/mL; ThermoFisher Scientific Cat# EN0531) and RNase T1 (200 units/ml; ThermoFisher Scientific, Cat# EN0542) mix, or with DNase 1 (10 units; Invitrogen™ TURBO™ DNase (2 U/μL), ThermoFisher Scientific, Cat# AM2238) or Benzonase (250 units; Sigma-Aldrich, Cat# E8263). Samples were ethanol-precipitated, resuspended, quantitated, slot-blotted and TOP3Bccs were detected with Anti-TOP3B antibody [EP7779]−C-terminal (Abcam, CAT#: ab183520).

### R-loop detection by Dot-Blot method using s9.6 Ab

For R-loop detection by slot-blot, genomic DNA was extracted from *TDRD3KO* HCT116 cells transfected with HA-TOP3B alone or together with TDRD3-FLAG constructs (as indicated) using DRIP protocol as described previously[12,54,55]. Briefly, cells were lysed in TE buffer containing SDS and proteinase K (at 37 °C overnight, Invitrogen™ Proteinase K Solution (20 mg/mL), RNA grade, ThermoFisher Scientific, Cat# 25530049), phase separated using phenol/chloroform/isoamyl alcohol (25:24:1), ethanol precipitated and resuspended in TE buffer. Genomic DNA was digested using cocktail of restriction enzymes (HindIII, SspI, EcoRI, BsrGI and XbaI; 30 U each) and again purified by phenol/chloroform/isoamyl alcohol (25:24:1) extraction. Increasing concentrations of genomic DNA were spotted on a nitrocellulose membrane, crosslinked with UV light (120 mJ/cm2)), blocked with PBS-Tween (0.1%) buffer and 5% non-fat milk (Room temperature for 1 hr) and incubated with mouse S9.6 antibody (1:500 dilution, overnight at 4 °C, Millipore Sigma, Cat# MABE1095). After washing with PBS-Tween (0.1%), membrane was incubated with HRP-conjugated anti-mouse secondary antibody, washed, and developed with ECL techniques. In case of RNase H treated control, 10 µg genomic DNA was pre-incubated with 20 U of RNase H (New England Biolabs, Cat# M0297L) for three hours at 37 °C.

### In vitro Biochemical Assay

The hairpin DNA oligo substrate with long 3′-tail: GGGATTATTG AACTGTTGTTCAAACTTTAGAACTAGCCATCCGATTTACACTTTGCCC CTATCCACCCC-3′FITC or the corresponding RNA oligo substrate: GGGAUUAUUGAACUGUUGUUCAAACUUUAGAACUAGCCAUCCGAU UUACACUUUGCCCCU-3′ Cy5 were synthesized by IDT. 75 nM of substrate was combined with 180 nM of purified recombinant TOP3B in 100 mM potassium glutamate (pH 7.0), 3 mM MgCl2, 0.02% v/v Tween-20, 1 mM DTT, with indicated amount of purified TDRD3 (45 nM, 90 nM, 180 nM, 360 and 720 nM) and incubated at 30 °C for 60 mins before addition of SDS (0.2%) to the samples to stop the reaction. The samples were resolved on 4-20% tris-glycine-SDS-PAGE and directedly visualized via the fluorescence of FITC or Cy5 on a typhoon scanner.

For dissociation kinetics of TOP3Bccs, 75 nM of substrate was combined with 180 nM of TOP3B in the presence or absence of 720 nM TDRD3 in the same buffer and incubated at 30 °C for 60 mins before addition of 2 volumes of reaction buffer to initiate dissociation. Aliquots were removed from the reaction and stopped with addition of SDS (0.2%) at indicated time. Samples were resolved on 4–20% tris-glycine-SDS-PAGE and directedly visualized via the fluorescence of FITC or Cy5 on a typhoon scanner.

### Statistics and reproducibility

Quantifications were carried out using the Fiji software[56]. Data are provided as means ± standard deviations (SD) from the number of independent experiments performed. Each experiments was performed three times independently with similar results unless otherwise mentioned in figure legends. Statistical analyses and graphical representation were carried out using GraphPad prism 10 software. Statistical test methods are described in each figure legend. Statistical significance is represented by ∗ and ∗∗ indicate a computed *p* value < 0.01 and <0.001 respectively. n.s. = not significant.

### Reporting summary

Further information on research design is available in the Nature Portfolio Reporting Summary linked to this article.

## Data availability

All unique/stable reagents generated in this study are available from the corresponding authors with a completed Materials Transfer Agreement. Original imaging data and raw data generated in this study are provided with this paper as a Source Data file. Any additional information required to reanalyze the data reported in this paper is available from the corresponding author upon request. Source data are provided with this paper.

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

## Acknowledgements

We thank Weidong Wang and Shuaikun Su (from NIA, NIH, Baltimore, MD, USA) for the TDRD3KO HCT116 cells. We thank Protein Expression Laboratory (Protein and Nucleic Acid Production—Center for Cancer Research [CCR]), NCI-Frederick, Maryland, for helping in the production of recombinant human TOP3B and TDRD3. Our studies are supported by the Center for Cancer Research, the Intramural Program of the National Cancer Institute, NIH (Z01-BC006161).

## Author contributions

Y.P. supervised the study. S.S. and Y.P. devised the concept. Y.P., S.S., and S.-Y.N.H. designed the experiments. S.S., X.Y., S.-Y.N.H., Y.S., L.K.S., P.K., and L.M.J. performed experiments and data analysis. S.S. and Y.P. wrote the manuscript.

## Competing interests

The authors declare no competing interests.
