## [Peer Review File · Nature Communications]

The TDRD3-USP9X complex and MIB1 regulate TOP3B homeostasis and prevent deleterious TOP3B cleavage complexesREVIEWER COMMENTS

Reviewer #1 (Remarks to the Author):

Saha et al. present an analysis of TOP3B regulation. They show that TOP3B levels are regulated by ubiquitination through a MIB1-USP9X axis mediated by direct interaction with TOP3B and TDRD3 respectively, and this axis stabilizes the TOP2B-TDRD3 complex over free TOP3B to minimize trapped TOP3Bccs and resulting DNA damage. Their study has implications for those interested in cellular mechanisms of gene regulation, genome integrity, targets for new anti-cancer therapeutics, and beyond. Overall the data is well presented, and the manuscript is clear and easy to follow. I suggest the following points for the authors to consider to strengthen their manuscript.

- 1) Figure 2d shows a very limited bit of information from a IP-mass spec experiment. The authors state that a number of DUBS were identified with USP9X being the highest, but this analysis is not shown. It would be helpful to include the list of hits and PSM values as a supplementary figure so the reader can interpret this experiment and understand how significant of a hit USP9X is. Same point for Fig. 2e. Later in the manuscript they co-IP the known TOP3B interactor MIB1 – was it present in the IP-mass spec? This would be an important control to demonstrate consistency in their experimental approaches.
- 2) Over/under expression of USP9X correlates with the levels of TOP3B. For example, in figure 3H it is shown that ectopic expression of USP9X protects TOP3B/TDRD3 from degradation, but it is not known whether this is due to protei-protein interactions or removal of ubiquitin chains. Ectopic expression of a catalytically inactive USP9X would be an important experiment to show that it is indeed due to removal of Ub rather than some other mechanism.
- 3) Fig 4g shows that ectopic expression of MIB1 increases TOP3B ubiquitination. A control experiment with a dominant negative or inactive mutant of MIB1 would support the author's conclusion this is due to MIB1 itself rather than altering other protein-protein interactions.
- 4) Fig 5d: what type of membrane is used? It is not described in the methods. Is it nylon and binding is mediated by the nucleic acid component, or PVDF/nitrocellulose and binding is mediated by the protein component?
- 5) Figure 6b: By visual inspection it appears that not all the TOP3B-DNA band intensities are affected equally by the addition of TDRD3. If the band intensities are quantified is this the case? If so, what can be inferred about the cellular role of TDRD3 if it differentially regulated TOP3B at different cleavage sites?
- 6) It seems that the role of the MIB1-USP9X-TDRD3 axis could be to equalize the levels of TOP3B and TDRD3, such that free TOP3B is removed and only TOP3B-TDRD3 is present in the cell. This would appear to be advantageous to the cell since free TOP3B seems to be a more deleterious enzyme. Does the author's data support this idea? Perhaps this point could be covered in the discussion section.

Reviewer #2 (Remarks to the Author):

Unlike most topoisomerases, TOP3B is stabilised by its partner TDRD3, but the physiological significance of this regulation and the key enzymes involved are poorly understood. In this paper, the Pommier lab used a proteomic approach to identify USP9X as the DUB that de-ubiquitinates TOP3B. Genetic epistasis experiments suggest that TDRD3 recruits USP9X to maintain optimal level of TOP3B proteostasis. The authors also identify MIB1 as the main E3 ubiquitin ligase that ubiquitylates TOP3B. The authors report that loss of TDRD3 increases TOP3Bcc in DNA and RNA, R-loops and DSBs, potentially providing an explanation for why the level of TOP3B need to be tightly controlled. Overall, the manuscript is well written, and data support the general conclusions. The following points need to be addressed to back-up existing data.

1. In Figure 1 the authors show that depletion or ectopic overexpression of TDRD3 controls the protein level of TOP3B in HEK293 and TDRD3KO HCT116 cells. Although unlikely, it is worth ruling out effects on transcription by measuring TOP3B transcript levels in this system.
2. The data nicely show the role of USP9X in regulating the ubiquitination of TOP3B. What is the ubiquitin chain that is implicated in this regulation, is it only K48?
3. In Figure 2 The USP9X-TOP3B interaction was significantly reduced in siTDRD3-transfected HCT116 cells, but not completely abrogated presumably because of residual expression of TDRD3. Quantification and statistical analyses should be added to this figure.
4. My main concern is the conclusion that TDRD3 recruits USP9X to TOP3B to stabilize TOP3B by deubiquitylating TOP3B. The conclusion is mainly based on measuring TOP3B ubiquitination in genetic epistasis analyses. What is the domain of TDRD3 that interacts with USP9X? Can the authors map the interaction domain and generate a TDRD3 mutant that is unable to bind USP9X to confirm the model?
5. In Figure 5 loss of TDRD3 led to a detectable interaction of TOP3B with PARP1, XRCC1 and LIG3. Is this interaction caused by the increased demand to repair TOP3Bcc and the resulting DSBs? Do these breaks occur in protein-coding genes or at promoters and enhancer where the majority of R-loops tend to accumulate?
6. Does TOP3B also interacts with other components required for repair, such as TDP2 and NuMA, in the absence of TRD3?
7. The increased formation of TOP3Bcc upon TDRD3 loss provides a nice explanation for the need to tightly regulate its level, however one would expect that TDP2 should be upregulated to help resolving the increased load of TOP3Bcc. Is that the case?
8. The data in Figure 6 shows TDRD3 stimulates the reversal / resealing of TOP3BBcc. How does TDRD3 perform this function. Addressing this point is critical in the understanding of the physiological significance of TOP3B/TDRD3 interaction, as unravelling the DUB and E3 ligase alone does not fully explain the obligate partnering between TOP3B and TRDR3.
9. TOP3B also interacts with other DUBs such as USP7 and USP39. It is worth using these DUBs as controls in the ubiquitination experiments, particularly those related to the conclusion regarding recruitment. i.e., does TDRD3 also interact with and recruit USP7 and USP39?
10. TOP3B ubiquitination is regulated by two E3 ligases; TRIM41 and MIB1, depending on whether they are chromatin bound in the form of TOP3Bcc or free. Does the presence of TDRD3 act as a molecular switch to favour MIB1-mediated ubiquitination over TRIM41-mediated ubiquitination? In the absence of TDRD3, does TRIM41 become the predominant E3 ligase to combat the increased TOP3Bcc levels? Addressing this question is critical to back-up the model born from this study.
11. Why similar obligate partners do not exist for the majority of other topoisomerases? Are TOP3Bcc more cytotoxic than say TOP1cc hence cells co-evolved a physically-coupled ubiquitination mechanism to regulate TOP3B steady state levels, and not TOP1? ...or are RNA-protein crosslinks more deleterious than DNA-protein crosslinks?

Minor comment:

In the discussion line 310, I think the authors meant USP9X may not be the only DUB (not the only E3 ligase).

Reviewer #3 (Remarks to the Author):

In this manuscript, the authors investigated the ubiquitination/deubiquitination pathway that controls the protein homeostasis of the TDRD3/TOP3B protein complex. They showed that USP9X is a major deubiquitinase that protects TDRD3 and TOP3B from ubiquitination and proteasome degradation. TDRD3 functions as a mediator for TOP3B-USP9X interaction. They also identified MIB1 as an E3 ubiquitin ligase that mediates TOP3B ubiquitination and degradation. In this case, TDRD3, however, is not regulated by MIB1. Thus, MIB1-mediated ubiquitination and USP9X-mediated deubiquitination

(bridged by TDRD3) could be the molecular mechanism underlying the regulation of TOP3B protein stability. To put this finding under cellular context, the authors showed that when cells are knocked out of TDRD3, TOP3B cleavage complex (TOP3Bcc) will accumulate and TOP3B's function in managing DNA topological problem is dampened, which results in elevated R-loops and DNA damage. So, degrading TOP3B, especially TOP3Bcc, when TDRD3 is absent is necessary for avoiding DNA damage. The authors also showed that TDRD3 can stimulate the turnover of TOP3Bcc in vitro, supporting other previous reports that TDRD3 can enhance the processivity of TOP3B on ssDNA and RNA.

This is a simple and logical study. The manuscript is well-written and easy to follow. USP9X has been reported as a binding partner of the TDRD3/TOP3B complex and its deubiquitinase activity stabilizes TDRD3. Thus, its role in stabilizing TOP3B is not unexpected. Identification of MIB1 as an E3 ligase of TOP3B is new, but how to differentiate its role from TRIM41, another E3 ligase of TOP3B they previously reported, is not addressed. Accumulation of TOP3Bcc in TDRD3 KO cells is novel because this could potentially explain some of the differences between TDRD3 KO and TOP3B KO. A few more experiments are needed to solidify the conclusion. There is also a little bit of disconnection between the identified UB/DUB pathway and TOP3Bcc regulation. The detailed critiques are:

1. More work needs to be done to more systematically compare MIB1 vs. TRIM41 catalyzed TOP3B ubiquitination. a) for MIB1, although knockdown and overexpression did affect TOP3B ubiquitination accordingly, these effects could be indirect. In vitro ubiquitination assay using recombinant proteins should be performed. The E3 ligase catalytic-deficient mutant should be included in both in vitro and in vivo ubiquitination assays; b) what type of ubiquitin linkage is catalyzed by MIB1? They have previously characterized TRIM41 linkage. Does the Ub linkage differ between free TOP3B vs TOP3Bcc? c) several other experiments could also help differentiate MIB1 vs. TRIM41. For example, mapping the interaction domains between TOP3B and these two E3 ligases may help understand if MIB1 prefers free TOP3B whereas TRIM41 prefers TOP3Bcc. Can the authors perform pull-down or CoIP experiment to see if different fractions of TOP3B (free vs TOP3Bcc) interact with different E3 ligases? Additionally, what are the ubiquitination sites catalyzed by MIB1 vs TRIM41? If identified, they should be able to use different K-R mutants to differentiate the impact of these two enzymes on TOP3B.
2. How is TOP3B ubiquitination regulated? It was shown before that CPT and Pla-B induce TOP3Bcc accumulation, which potentially activates TOP3B ubiquitination. Is this process leading to TOP3B ubiquitination by TRIM41 or MIB1 or both?
3. Regarding TOP3Bcc accumulation in cells without TDRD3. This is an interesting finding, but more experiments need to be performed to support the conclusion. a) can they detect TOP3Bcc in TDRD3 KO cells? The results shown in Fig. 5 and Supp Fig. 1 are all done by using overexpression, which could introduce some artifacts (overexpression of TDRD3 could cause stress granule formation). If they think low TOP3B level caused by UB/proteasome degradation upon TDRD3 loss is an issue, they can treat cells with MG132 or knockdown MIB1, TRIM41 individually, or both. What happens with USP9X overexpression or knockdown? These experiments could also help establish more connectivity between the first part and second part of the story. b) which domain of TDRD3 is required for reducing TOP3Bcc? The authors should include several functional domain mutations of TDRD3 to see if the N-terminal OB-fold, UBA domain, or the Tudor domain are involved in this regulation.
4. In Fig. 6, the overall effect of TDRD3 in stimulating TOP3B's activity on RNA is marginal. It does not seem to be dosage responsive. The only obvious change is the last lane (Fig. 6B), but it might just be the loading issue. Also, the OB-fold of TDRD3 is mainly involved in TOP3B interaction and could also interact with ssDNA. An OB-fold deletion mutant should be included in both assays. Another important control is the TOP3B R338W that forms strong TOP3Bcc. Will TDRD3 be able to stimulate its activity?

Reviewer #1 (Remarks to the Author):

Saha et al. present an analysis of TOP3B regulation. They show that TOP3B levels are regulated by ubiquitination through a MIB1-USP9X axis mediated by direct interaction with TOP3B and TDRD3 respectively, and this axis stabilizes the TOP3B-TDRD3 complex over free TOP3B to minimize trapped TOP3Bccs and resulting DNA damage. Their study has implications for those interested in cellular mechanisms of gene regulation, genome integrity, targets for new anti-cancer therapeutics, and beyond. Overall, the data is well presented, and the manuscript is clear and easy to follow. I suggest the following points for the authors to consider to strengthen their manuscript.

Answer: Thank you for your constructive comments and valuable suggestions to strengthen our manuscript. Indeed, our study provides new implications connecting not only TDRD3-USP9X complex and TOP3B but also extends the relevance of MIB1 as a regulator of TOP3B cellular function.

1) Figure 2d shows a very limited bit of information from a IP-mass spec experiment. The authors state that a number of DUBS were identified with USP9X being the highest, but this analysis is not shown. It would be helpful to include the list of hits and PSM values as a supplementary figure so the reader can interpret this experiment and understand how significant of a hit USP9X is. Same point for Fig. 2e. Later in the manuscript they co-IP the known TOP3B interactor MIB1 – was it present in the IP-mass spec? This would be an important control to demonstrate consistency in their experimental approaches.

Answer: Part 1: As suggested, we have now listed all the DUBs identified in two IP-mass spec experiments (with PSM values > 3) in Fig. 2d and 2e. Part 2: TOP3B was reported as a potential interactor of MIB1 in studies listing Mass Spec interactors of MIB1 but not focusing on TOP3B (Dho et al., 2019; Wang et al., 2013). MIB1 was also present in our IP-Mass spec, but PSM value was not very high. We confirmed the TOP3B-MIB1 interaction by performing TOP3B and MIB1 IP-Western blot experiments (Fig. 4a and 4b). This has been included in our revision.

2) Over/under expression of USP9X correlates with the levels of TOP3B. For example, in figure 3H it is shown that ectopic expression of USP9X protects TOP3B/TDRD3 from degradation, but it is not known whether this is due to protein-protein interactions or removal of ubiquitin chains. Ectopic expression of a catalytically inactive USP9X would be an important experiment to show that it is indeed due to removal of Ub rather than some other mechanism.

Answer: To address this point, we generated catalytically inactive USP9X (USP9X C1566S mutant), which we overexpressed in HEK293 cells along with the USP9X wild-type counterpart. Our new data show that overexpression of wild-type USP9X stabilizes TOP3B and TDRD3 whereas the USP9X C1566S mutant fails to stabilize TOP3B and TDRD3 (Supplementary Fig. 2c). This demonstrates that the catalytic activity of USP9X is critical to remove Ub from TOP3B and to stabilize cellular TOP3B. Thank you.

3) Fig 4g shows that ectopic expression of MIB1 increases TOP3B ubiquitination. A control experiment with a dominant negative or inactive mutant of MIB1 would support the author's conclusion this is due to MIB1 itself rather than altering other protein-protein interactions.

Answer: As suggested we generated catalytically inactive MIB1 (MIB1 C985S active site mutant), which we over-expressed in HEK293 cells along with the MIB1 wild-type counterpart. Overexpression of wild-type MIB1 increased ubiquitination of TOP3B, decreased total cellular TOP3B level whereas MIB1 C985S mutant failed to affect TOP3B protein ubiquitination or stability (Supplementary Fig. 3c). The manuscript has been revised accordingly. Thank you.

4) Fig 5d: what type of membrane is used? It is not described in the methods. Is it nylon and binding is mediated by the nucleic acid component, or PVDF/nitrocellulose and binding is mediated by the protein component?

Answer: For RADAR assay we used nitrocellulose membrane (Bio-Rad, 0.45 μ m, Catalog# 1620115) which can be used for transfer of both proteins and nucleic acids (can immobilize both proteins and nucleic acids) (Kiianitsa and Maizels, 2013; Anand et al., 2017; Meroni and Vindigni, 2022; Sun 2023). These points have been clarified in our revised manuscript. Thank you.

5) Figure 6b: By visual inspection it appears that not all the TOP3B-DNA band intensities are affected equally by the addition of TDRD3. If the band intensities are quantified is this the case? If so, what can be inferred about the cellular role of TDRD3 if it differentially regulated TOP3B at different cleavage sites?

Answer: The difference in the molecular weight of TOP3Bccs formed at different TOP3B cleavage sites on the DNA substrate [the 69 nucleotide long oligo used in this assay has 3 different TOP3B cleavage sites at 17, 29 and 55 nucleotides from the 5'-end (Figure S3B, Saha et al., 2020, Cell Rep)] is expected to be below the resolution power of the gel. The nature of the higher molecular weight TOP3Bcc species has been a challenging point to address since our first publication reporting this in vitro DNA TOP3Bcc formation assay (Saha et al., 2020 Cell Rep, Wang et al., 2023, PNAS). We believe the high molecular weight TOP3Bcc species likely represent a higher order TOP3Bccs associated with more complex nucleic acid structures, and as such, could potentially restrict physical or structural accessibility for potential interactions with TDRD3, which is a reasonable assumption since TDRD3 is known to physically interact with TOP3B (Xu et al., 2013; Stoll et al., 2013). Addressing the exact nature of the shifted species will require further detailed studies and is beyond the scope of the current report.

6) It seems that the role of the MIB1-USP9X-TDRD3 axis could be to equalize the levels of TOP3B and TDRD3, such that free TOP3B is removed and only TOP3B-TDRD3 is present in the cell. This would appear to be advantageous to the cell since free TOP3B seems to be a more deleterious enzyme. Does the author's data support this idea? Perhaps this point could be covered in the discussion section.

Answer: We appreciate your insightful comment, and share your interpretation. Indeed, from our biochemical and cellular studies, free TOP3B is deleterious for genome integrity, and the MIB1-TDRD3-USP9X axis appears to work in an orchestrated manner to maintain stoichiometric levels of TOP3B and TDRD3 in such way that free TOP3B is removed and only TOP3B-TDRD3 is present in the cell. We now have included this interpretation in the Discussion section, and our proposed model has been updated in Fig. 7. Thank you.

Reviewer #2 (Remarks to the Author):

Unlike most topoisomerases, TOP3B is stabilised by its partner TDRD3, but the physiological significance of this regulation and the key enzymes involved are poorly understood. In this paper, the Pommier lab used a proteomic approach to identify USP9X as the DUB that de-ubiquitinates TOP3B. Genetic epistasis experiments suggest that TDRD3 recruits USP9X to maintain optimal level of TOP3B proteostasis. The authors also identify MIB1 as the main E3 ubiquitin ligase that ubiquitylates TOP3B. The authors report that loss of TDRD3 increases TOP3Bcc in DNA and RNA, R-loops and DSBs, potentially providing an explanation for why the level of TOP3B need to be tightly controlled. Overall, the manuscript is well written, and data support the general conclusions. The following points need to be addressed to back-up existing data.

Answer: Thank you for your positive remarks and valuable inputs to improve our manuscript.

1. In Figure 1 the authors show that depletion or ectopic overexpression of TDRD3 controls the protein level of TOP3B in HEK293 and TDRD3KO HCT116 cells. Although unlikely, it is worth ruling out effects on transcription by measuring TOP3B transcript levels in this system.

Answer: As suggested we performed qRT-PCR to measure TOP3B transcripts in siTDRD3-transfected HEK293 cells and *TDRD3KO* HCT116 cells. In both conditions TDRD3 down regulation did affect cellular TOP3B RNA levels compared to control siRNA transfected or wild-type cells (Supplementary Fig. 1a and 1b).

2. The data nicely show the role of USP9X in regulating the ubiquitination of TOP3B. What is the ubiquitin chain that is implicated in this regulation, is it only K48?

Answer: To determine the ubiquitin chains formed on TOP3B we transfected *TDRD3KO* HCT116 cells with either wild-type HA-tagged ubiquitin (Ub) or HA-tagged lysine-to-arginine Ub mutants for each of the 7 lysine residues (K6R-Ub, K11R-Ub, K27R-Ub, K29R-Ub, K33R-Ub, K48R-Ub, and K63R-Ub) of ubiquitin. Pulling-down TOP3B and probing with ubiquitin antibody showed both K11- and K48-linked ubiquitin chains (Supplementary Fig. 1c), which is consistent with the proteasomal degradation of free TOP3B in the absence of TDRD3. To determine whether USP9X can process the K11- and K48-linked ubiquitin chains formed on TOP3B, we pulled down TOP3B from HEK293 cells after siRNA-mediated USP9X downregulation or after ectopic expression of USP9X-FLAG and probed with K11- and K48-specific ubiquitin antibodies. USP9X removed both K11- and K48-linked ubiquitin chains from free TOP3B. These new results are included in Supplementary Fig. 2a-b. Thank you.

3. In Figure 2 The USP9X-TOP3B interaction was significantly reduced in siTDRD3-transfected HCT116 cells, but not completely abrogated presumably because of residual expression of TDRD3. Quantification and statistical analyses should be added to this figure.

Answer: As suggested we have added quantification and statistical analyses in Fig. 2g. Thank you.

4. My main concern is the conclusion that TDRD3 recruits USP9X to TOP3B to stabilize TOP3B by deubiquitylating TOP3B. The conclusion is mainly based on measuring TOP3B ubiquitination

in genetic epistasis analyses. What is the domain of TDRD3 that interacts with USP9X? Can the authors map the interaction domain and generate a TDRD3 mutant that is unable to bind USP9X to confirm the model?

Answer: An Earlier Study (Narayanan et al., 2017, Cell Discovery) showed that the C-terminal domain of TDRD3 interacts with the USP9X C-terminal domain. To find out whether lack of interaction between TDRD3-USP9X affects TOP3B stability, we generated a C-terminal deletion mutant of TDRD3 (1-352 TDRD3-FLAG), which we overexpressed in *TDRD3KO* HCT116 cells along with TDRD3 wild-type counterpart (Supplementary Fig. 1d). The C-terminal deletion mutant of TDRD3 (1-352 aa) could not stabilize TOP3B compared to wild-type TDRD3. These new results have been included in Supplementary Figure 1e. Thank you.

5. In Figure 5 loss of TDRD3 led to a detectable interaction of TOP3B with PARP1, XRCC1 and LIG3. Is this interaction caused by the increased demand to repair TOP3Bcc and the resulting DSBs? Do these breaks occur in protein-coding genes or at promoters and enhancer where the majority of R-loops tend to accumulate?

Answer: Indeed, increased interaction of TOP3B with PARP1, XRCC1 and LIG3 in absence of TDRD3 is likely caused by the increased demand to repair TOP3Bccs and the resulting SSBs/DSBs. WE agree that it will be interesting to map TOP3Bcc sites in cells in the presence and absence of TDRD3. However, locating the exact positions TOP3Bccs and SSBs/DSBs in TDR3-deficient cells is beyond the scope of the current study. As R-loops are more prone to form at promoters or transcription termination regions, it is highly possible that in absence of TDRD3, TOP3Bcc and the resulting SSBs/DSBs can form in these genomic regions. We have included this point in our revision.

6. Does TOP3B also interacts with other components required for repair, such as TDP2 and NuMA, in the absence of TDRD3?

Answer: In our IP-Mass spec experiments we did not retrieve TDP2 or NuMA in the absence of TDRD3. Further studies are warranted to find out whether TDP2, the proteasome and TRIM41 play any important roles for processing of TOP3Bccs formed in absence of TDRD3. Nevertheless we have included additional experiments in our revised manuscript to elucidate the differential roles of TRIM41 and MIB1 in TDRD3-deficient cells (please see answer to question 10).

7. The increased formation of TOP3Bcc upon TDRD3 loss provides a nice explanation for the need to tightly regulate its level, however one would expect that TDP2 should be upregulated to help resolving the increased load of TOP3Bcc. Is that the case?

Answer: As suggested we measured TDP2 levels in TDRD3-deficient cells. We did not see TDP2 upregulation after down-regulation of TDRD3 or in the absence of TDRD3. These new results have been included in our revised manuscript (Supplementary Fig. 6a and 6b). Existing literature also supports this as treatment of cells with TOP1/ TOP2 trapping agents (camptothecin and etoposide as poisons of topoisomerase 1/2 cleavage complexes) do not change TDP1/ TDP2 protein levels (Sun et al, 202, Nat Commun; Zhang et al., 2022, Nat Commun; Bian et al., 2016, Oncotarget; Dang et al., 2020, Cell Death Dis.).

8. The data in Figure 6 shows TDRD3 stimulates the reversal / resealing of TOP3BBcc. How does TDRD3 perform this function. Addressing this point is critical in the understanding of the physiological significance of TOP3B/TDRD3 interaction, as unravelling the DUB and E3 ligase alone does not fully explain the obligate partnering between TOP3B and TRDR3.

Answer: Given the fact that TDRD3 physically interacts with TOP3B (the DUF-OB-Fold of TDRD3 interacts with domain II of TOP3B, and the Tudor domain of TDRD3 interacts with TOP3B in a methylation-dependent manner; Xu et al., 2013; Stoll et al., 2013; Huang et al., 2018), it is not excluded that TDRD3 binding might favor TOP3B conformations making its catalytic intermediates (TOP3Bccs) more transient. We are pursuing our structural studies (Yang, X. et al., 2022) using purified TDRD3 and TOP3B to clarify this question.

9. TOP3B also interacts with other DUBs such as USP7 and USP39. It is worth using these DUBs as controls in the ubiquitination experiments, particularly those related to the conclusion regarding recruitment. i.e., does TDRD3 also interact with and recruit USP7 and USP39?

Answer: Indeed, we report USP7 and USP39 in our TOP3B interactome (Fig. 2d and 2e). Our IP-MS experiments for TOP3B (wild-type vs *TDRD3KO* HCT116) and TDRD3 (wild-type vs *TOP3BKO* HCT116) indicate that USP7, USP10 and USP39 can interact with TOP3B in the absence of TDRD3. The focus of the present study is on USP9X. Notably, we found that USP9X is recruited to TOP3B by TDRD3. Our report should stimulate further studies to determine the potential the potential differential roles of additional DUBs.

10. TOP3B ubiquitination is regulated by two E3 ligases; TRIM41 and MIB1, depending on whether they are chromatin bound in the form of TOP3Bcc or free. Does the presence of TDRD3 act as a molecular switch to favour MIB1-mediated ubiquitination over TRIM41-mediated ubiquitination? In the absence of TDRD3, does TRIM41 become the predominant E3 ligase to combat the increased TOP3Bcc levels? Addressing this question is critical to back-up the model born from this study.

Answer: Thank you for raising this important question. To address it, we pulled down free TOP3B (no nuclease/benzonase in IP buffer) and total cellular TOP3B (benzonase in IP buffer) from HEK293 cells and compared TRIM41 and MIB1. MIB1 interacted with both free and total cellular TOP3B whereas TRIM41 interacted only with total cellular fraction of TOP3B. These new data have been included in our revised manuscript. The new Supplementary Figures 7a and 7b show that TRIM41 interacts with TOP3B in the presence of nucleic acids. To further this conclusion, we isolated chromatin fractions from HEK293 cells and pulled down TOP3B. Only TRIM41 interacted with chromatin bound TOP3B (Supplementary Fig. 7c). To determine the specific roles of TRIM41 and MIB1 in regulating TOP3B levels, we also checked TOP3Bcc levels in FLAG-TOP3B-transfected *TDRD3KO* HCT116 cells after siRNA mediated down regulation of TRIM41 and MIB1 in the absence of TDRD3. Downregulation of both TRIM41 and MIB1 increased in TOP3Bcc level (new Supplementary Fig. 7d). To find out whether both MIB1 can ubiquitylate TOP3Bccs, we pulled down TOP3Bcc from RADAR assay samples prepared from FLAG-TOP3B-transfected *TDRD3KO* HCT116 cells after siRNA of TRIM41 and MIB1. Only TRIM41 down-regulation affected TOP3Bcc ubiquitination level (Supplementary Fig. 7e). This finding indicated that in absence of TDRD3, TRIM41 ubiquitylates TOP3Bccs while MIB1

ubiquitylates free TOP3B. Thank you for this interesting point, which is highlighted in our revised manuscript, both in the text and data figures and in the summary Figure 7.

11. Why similar obligate partners do not exist for the majority of other topoisomerases? Are TOP3Bcc more cytotoxic than say TOP1cc hence cells co-evolved a physically-coupled ubiquitination mechanism to regulate TOP3B steady state levels, and not TOP1? ...or are RNA-protein crosslinks more deleterious than DNA-protein crosslinks?

Answer: We previously showed that TOP1, TOP2A and TOP2B form complexes with the SUMO ligase PIAS4 in the absence of topoisomerase poisons (Sun et al., 2020, Science Advances). Once TOP1/2 are trapped on DNA, PIAS4 SUMOylates TOP1/2 and initiates the recruitment of the SUMO-targeted ubiquitin ligase RNF4 for proteasomal targeting of TOP1/2 cleavage complexes (Sun et al., 2020, Science Advances). Further studies are warranted to elucidate whether RNA-protein crosslinks are more deleterious than DNA-protein crosslinks.

Minor comment:

In the discussion line 310, I think the authors meant USP9X may not be the only DUB (not the only E3 ligase).

Answer: We corrected this typographical error. Thank you.

Reviewer #3 (Remarks to the Author):

In this manuscript, the authors investigated the ubiquitination/deubiquitination pathway that controls the protein homeostasis of the TDRD3/TOP3B protein complex. They showed that USP9X is a major deubiquitinase that protects TDRD3 and TOP3B from ubiquitination and proteasome degradation. TDRD3 functions as a mediator for TOP3B-USP9X interaction. They also identified MIB1 as an E3 ubiquitin ligase that mediates TOP3B ubiquitination and degradation. In this case, TDRD3, however, is not regulated by MIB1. Thus, MIB1-mediated ubiquitination and USP9X-mediated deubiquitination (bridged by TDRD3) could be the molecular mechanism underlying the regulation of TOP3B protein stability. To put this finding under cellular context, the authors showed that when cells are knocked out of TDRD3, TOP3B cleavage complex (TOP3Bcc) will accumulate and TOP3B's function in managing DNA topological problem is dampened, which results in elevated R-loops and DNA damage. So, degrading TOP3B, especially TOP3Bcc, when TDRD3 is absent is necessary for avoiding DNA damage. The authors also showed that TDRD3 can stimulate the turnover of TOP3Bcc in vitro, supporting other previous reports that TDRD3 can enhance the processivity of TOP3B on ssDNA and RNA. This is a simple and logical study. The manuscript is well-written and easy to follow. USP9X has been reported as a binding partner of the TDRD3/TOP3B complex and its deubiquitinase activity stabilizes TDRD3. Thus, its role in stabilizing TOP3B is not unexpected. Identification of MIB1 as an E3 ligase of TOP3B is new, but how to differentiate its role from TRIM41, another E3 ligase of TOP3B they previously reported, is not addressed. Accumulation of TOP3Bcc in TDRD3 KO cells is novel because this could potentially explain some of the differences between TDRD3 KO and TOP3B KO. A few more experiments are needed to solidify the conclusion. There is also a little bit of disconnection between the identified UB/DUB pathway and TOP3Bcc regulation. The detailed critiques are:

Answer: Thank you for finding the study logical and easy to follow, and the manuscript well-written. Thank you for the constructive suggestions.

1. More work needs to be done to more systematically compare MIB1 vs. TRIM41 catalyzed TOP3B ubiquitination. a) for MIB1, although knockdown and overexpression did affect TOP3B ubiquitination accordingly, these effects could be indirect. In vitro ubiquitination assay using recombinant proteins should be performed. The E3 ligase catalytic-deficient mutant should be included in both in vitro and in vivo ubiquitination assays; b) what type of ubiquitin linkage is catalyzed by MIB1? They have previously characterized TRIM41 linkage. Does the Ub linkage different between free TOP3B vs TOP3Bcc? c) several other experiments could also help differentiate MIB1 vs. TRIM41. For example, mapping the interaction domains between TOP3B and these two E3 ligases may help understand if MIB1 prefers free TOP3B whereas TRIM41 prefers TOP3Bcc. Can the authors perform pull-down or CoIP experiment to see if different fractions of TOP3B (free vs TOP3Bcc) interact with different E3 ligases? Additionally, what are the ubiquitination sites catalyzed by MIB1 vs TRIM41? If identified, they should be able to use different K-R mutants to differentiate the impact of these two enzymes on TOP3B.

Answer: We agree that comparing the functions of MIB1 and TRIM41 in TOP3B ubiquitination is important and relevant for our study. a) As suggested, we generated catalytically inactive MIB1 (MIB1 C985S active site mutant), which we over-expressed in HEK293 cells in comparison with the MIB1 wild-type counterpart. Overexpression of wild-type MIB1 increased ubiquitination of TOP3B, decreased total cellular TOP3B level whereas MIB1 C985S mutant

failed to affect TOP3B protein ubiquitination or stability (new Supplementary Fig. 3c). We could not perform *in vitro* ubiquitination assay because of unavailability of purified MIB1 protein. b) Yet, we have been able to determine the polyubiquitin linkages of free TOP3B in the absence of TDRD3. For that, we transfected *TDRD3KO* HCT116 cells with either wild-type HA-tagged ubiquitin (Ub) or HA-tagged lysine-to-arginine Ub mutants for each of the 7 lysine residues involved in polyubiquitin chains (K6R-Ub, K11R-Ub, K27R-Ub, K29R-Ub, K33R-Ub, K48R-Ub, and K63R-Ub), pulled down TOP3B and probed with ubiquitin antibody. Both K11- and K48-linked ubiquitin chains were observed on TOP3B (new Supplementary Fig. 1c), which is consistent with the proteasomal degradation of free TOP3B in the absence of TDRD3. To determine whether MIB1 can generate K11- and K48-linked ubiquitin chains, we pulled down TOP3B from *TDRD3KO* HCT116 cells after siRNA mediated MIB1 downregulation or after ectopic expression of MIB1-FLAG and probed with K11- and K48-specific ubiquitin antibodies. The new Supplementary Figure 3 (panels a-b) shows that MIB1 forms both K11- and K48-linked ubiquitin chains on free TOP3B. c) To address the specific roles of the two TOP3B E3 ligases (TRIM41 and MIB1), we pulled down free TOP3B (no nuclease/benzonase in IP buffer) and total cellular TOP3B (benzonase in IP buffer) from HEK293 cells and looked for TRIM41 and MIB1 interaction. MIB1 interacted with both free and total cellular TOP3B whereas TRIM41 interacted only with total cellular TOP3B (new Supplementary Fig. 7a and 7b). This indicates that TRIM41 interacts with TOP3Bccs, i.e. in the presence of nucleic acids. To confirm this possibility, we isolated chromatin fractions from HEK293 cells and pulled down TOP3B. We found that only TRIM41 interacts with chromatin bound TOP3B (new Supplementary Fig. 7c). To further establish the specific roles of TRIM41 and MIB1 in regulating TOP3Bccs in the absence of TDRD3, we checked TOP3Bcc levels in FLAG-TOP3B-transfected *TDRD3KO* HCT116 cells after siRNA mediated down regulation of TRIM41 and MIB1. Downregulation of both TRIM41 and MIB1 caused increase in TOP3Bcc level (new Supplementary Fig. 7d). We also pulled down TOP3Bcc from RADAR assay samples prepared from FLAG-TOP3B transfected *TDRD3KO* HCT116 cells after siRNA mediated down regulation of TRIM41 and MIB1. Only TRIM41 down regulation affected TOP3Bcc ubiquitination level (new Supplementary Fig. 7e). This finding indicates that, in the absence of TDRD3, TRIM41 ubiquitylates TOP3Bccs whereas MIB1 ubiquitylates free TOP3B. These new experiments have been included in our revised manuscript and the differential roles of MIB1 (E3 for free TOP3B) and TRIM41 (E3 for TOP3B-DNA protein crosslinks) have been schematized in our revised summary figure (Fig. 7). Thank you.

2. How is TOP3B ubiquitination regulated? It was shown before that CPT and Pla-B induce TOP3Bcc accumulation, which potentially activates TOP3B ubiquitination. Is this process leading to TOP3B ubiquitination by TRIM41 or MIB1 or both?

Answer: We have shown previously that treatment with CPT or Pla-B recruits TOP3B to R-loops as indicated by the accumulation of TOP3B catalytic intermediates (TOP3Bccs) (Saha et al., 2022, Cell reports). Although we have not shown that these conditions induce TOP3B ubiquitination, when we transfect our self-trapping TOP3B mutant (R338W TOP3B), we find TOP3Bcc ubiquitination catalyzed by TRIM41. Additionally, as discussed above (answer to question 1c), MIB1 down regulation does not affect TOP3Bcc ubiquitination level as compared to TRIM41 downregulation. These points have been included in our revised manuscript and in the new Supplementary Figure 7e.

3. Regarding TOP3Bcc accumulation in cells without TDRD3. This is an interesting finding, but more experiments need to be performed to support the conclusion. a) can they detect TOP3Bcc in TDRD3 KO cells? The results shown in Fig. 5 and Supp Fig. 1 are all done by using overexpression, which could introduce some artifacts (overexpression of TDRD3 could cause stress granule formation). If they think low TOP3B level caused by UB/proteasome degradation upon TDRD3 loss is an issue, they can treat cells with MG132 or knockdown MIB1, TRIM41 individually, or both. What happens with USP9X overexpression or knockdown? These experiments could also help establish more connectivity between the first part and second part of the story. b) which domain of TDRD3 is required for reducing TOP3Bcc? The authors should include several functional domain mutations of TDRD3 to see if the N-terminal OB-fold, UBA domain, or the Tudor domain are involved in this regulation.

Answer: Thank you for your interest in our finding that TOP3Bccs accumulate in cells without TDRD3. a) As suggested, to study the effect of TDRD3 depletion on endogenous TOP3Bccs, we used our recently reported TOP3B trapping drugs NSC96932 and NSC690634 (Wang et al., 2023, PNAS). Wild-type and *TDRD3KO* HCT116 cells were treated with MG132 (10 μ M, 3 h) and NSC96932 (100 μ M, 1 h) or NSC690634 (100 μ M, 1 h) and RADAR assays were performed to detect endogenous TOP3Bcc. We found that *TDRD3KO* HCT116 cells accumulate more endogenous TOP3Bcc compared to their wild-type counterpart (new Supplementary Fig. 4a and 4b). These findings corroborate our results in *TDRD3KO* cells transfected with the HA-TOP3B construct alone or in combination with FLAG-tagged TDRD3 construct (Fig. 5a). To detect the accumulation of endogenous TOP3Bccs in *TDRD3KO* HCT116 cells, we also performed immunoprecipitation experiments using TOP3B antibody with RADAR assay samples from MG132-treated wild-type and *TDRD3KO* HCT116 cells. Accumulation of endogenous TOP3Bccs could be detected in *TDRD3KO* HCT116 cells. These results are included in the new Supplementary Figure 4c. b) To address which domain of TDRD3 regulates TOP3B levels, we co-transfected *TDRD3KO* HCT116 cells with HA-TOP3B and/or five different TDRD3 variants: full length TDRD3 (744 aa), TDRD3 transcript isoform 2 having 93 amino acids deletions in N-terminal (651 aa), 1-352 TDRD3-FLAG (with partially deleted linker domain and deleted Tudor domain, FMRP interaction domain and exon junction binding motif) and 1-588 TDRD3-FLAG (with deleted Tudor domain, FMRP interaction domain and exon junction binding motif) and 1-171 TDRD3-MBP (shown in Supplementary Fig. 4d) and RADAR assays were performed to detect TOP3Bccs. Transfection with the 1-352 TDRD3-FLAG and 1-588 TDRD3-FLAG mutants (having deletions in Tudor domain) lowered TOP3Bcc in *TDRD3KO* HCT116 cells more or less with similar efficiency as full length TDRD3 (1-744 aa) (shown in new Supplementary Fig. 4e). In contrast, TDRD3 transcript isoform 2 having partial deletions in N-terminal DUF-OB fold (first 93 amino acids; 651 aa) and 1-171 TDRD3-MBP having deletions in UBA domain could partially reduce TOP3Bcc in *TDRD3KO* HCT116 cells (shown in Supplementary Fig. 4e). These results suggest that the N-terminus of TDRD3 (which contains DUF-OB fold, UBA domain and linker domain) is predominantly involved in regulating cellular TOP3Bcc levels. Thank you.

4. In Fig. 6, the overall effect of TDRD3 in stimulating TOP3B's activity on RNA is marginal. It does not seem to be dosage responsive. The only obvious change is the last lane (Fig. 6B), but it might just be the loading issue. Also, the OB-fold of TDRD3 is mainly involved in TOP3B interaction and could also interact with ssDNA. An OB-fold deletion mutant should be included

in both assays. Another important control is the TOP3B R338W that forms strong TOP3Bcc. Will TDRD3 be able to stimulate its activity?

Answer: Thank you for raising these points. In our *in vitro* assay, TDRD3 has a more striking effect on DNA-TOP3Bccs than RNA-TOP3Bccs. We found that a 4-fold excess of TDRD3 is required to visibly reduce the RNA-TOP3Bcc levels, which is corroborated by the reversal gel in figure 6d showing that it is not due to loading differences. Importantly, we note that the TOP3Bccs forms more readily on DNA than RNA, and the levels of RNA-TOP3Bccs in the sample is low. Therefore it is likely that the reduced effect of TDRD3 on RNA-TOP3Bccs can be attributed to the lower level of RNA-TOP3Bccs, so that the high level of free TOP3B in the sample is competing with RNA-TOP3Bccs for binding with TDRD3. Given the fact that TDRD3 has been shown to directly interact with TOP3B (DUF-OB Fold of TDRD3 interacts with domain II of TOP3B and Tudor domain of TDRD3 interacts with TOP3B in a methylation-dependent manner; Xu et al., 2013; Stoll et al., 2013; Huang et al., 2018), it is tempting to speculate that TDRD3 binding to TOP3B might favor conformations of TOP3B which facilitate the catalytic turnover TOP3B, thereby reducing detectable TOP3Bccs. Further detailed mechanistic/structural studies are warranted to decipher how TDRD3 differentially stimulates the reversal / resealing of TOP3BBcc in DNA and RNA and we are carrying out structural studies with purified TDRD3 and TOP3B to address this question. Thank you.

REVIEWERS' COMMENTS

Reviewer #1 (Remarks to the Author):

The revised manuscript submitted by Saha and colleagues contains additional data, analyses, and control experiments that strengthens and supports their conclusions. I have no further comments, and I look forward to seeing it in press!

Reviewer #2 (Remarks to the Author):

The authors have thoroughly addressed all my comments.

Reviewer #3 (Remarks to the Author):

In the revised manuscript, the authors performed additional experiments to address the reviewer's critiques. These new results provided solid support to the working model. I appreciate the efforts and most of my comments have been addressed. There remain a few points that might have been missed by the authors, but I think they are important to further strengthen the manuscript.

1. In supp Figure 4e, the expression levels of different TDRD3 truncations need to be considered when comparing their activity in reducing TOP3Bcc. The authors need to exclude the possibility that the differences in the slot-blot signals might be due to the differential protein levels. The amount of TOP3Bcc should be normalized and quantified to the amount of each truncation being expressed.
2. As I have mentioned before, there are two important controls that should be included to address TDRD3's stimulating effect on TOP3B activity (Fig. 6). 1) TDRD3 isoform 2, which does not interact with TOP3B and shows least activity in reducing TOP3Bcc (as shown in their new supp Figure 4e), and 2) TOP3B R338W that forms strong TOP3Bcc, but likely non-regulatable by TDRD3. These two controls will strengthen the conclusion.
3. On page 4. line 85, Fig. 1e-f should be Fig. 1e & 1g; and Fig. 1g-h should be Fig. 1f & 1h.
4. All the IP-mass spec results should be included as supplementary tables.

Reviewer #1 (Remarks to the Author):

The revised manuscript submitted by Saha and colleagues contains additional data, analyses, and control experiments that strengthens and supports their conclusions. I have no further comments, and I look forward to seeing it in press!

Answer: Thank you so much for your constructive comments and valuable suggestions to strengthen our manuscript.

Reviewer #2 (Remarks to the Author):

The authors have thoroughly addressed all my comments.

Answer: Thank you.

Reviewer #3 (Remarks to the Author):

In the revised manuscript, the authors performed additional experiments to address the reviewer's critiques. These new results provided solid support to the working model. I appreciate the efforts and most of my comments have been addressed. There remain a few points that might have been missed by the authors, but I think they are important to further strengthen the manuscript.

Answer: Thank you for your positive remarks and valuable inputs to improve our manuscript.

1. In supp Figure 4e, the expression levels of different TDRD3 truncations need to be considered when comparing their activity in reducing TOP3Bcc. The authors need to exclude the possibility that the differences in the slot-blot signals might be due to the differential protein levels. The amount of TOP3Bcc should be normalized and quantified to the amount of each truncation being expressed.

Answer: As suggested, we have now included western blot experiment to show that different variants of TDRD3 had comparable expression levels (New Supplementary Fig. 4e). We also added quantitations for TOP3Bcc levels in cells after transfection with different TDRD3 variants (New Supplementary Fig. 4g). Thank you.

2. As I have mentioned before, there are two important controls that should be included to address TDRD3's stimulating effect on TOP3B activity (Fig. 6). 1) TDRD3 isoform 2, which does not interact with TOP3B and shows least activity in reducing TOP3Bcc (as shown in their new supp Figure 4e), and 2) TOP3B R338W that forms strong TOP3Bcc, but likely non-regulatable by TDRD3. These two controls will strengthen the conclusion.

Answer: Thank you for raising these points and we agree that performing these two control experiments will further strengthen the conclusions. Yet, we could not perform these biochemical experiments as we could not purify these two proteins (TDRD3 isoform 2 and R338W TOP3B) on time. But from the cellular study and biochemical experiments with DNA and RNA substrate already included with the manuscript, it is clear that TDRD3 stimulates the turnover of TOP3Bccs.

Further detailed mechanistic/structural studies are ongoing in our lab to address these questions.
Thank you.

3. On page 4. line 85, Fig. 1e-f should be Fig. 1e & 1g; and Fig. 1g-h should be Fig. 1f & 1h.

Answer: We corrected this typographical error. Thank you.

4. All the IP-mass spec results should be included as supplementary tables.

Answer: All the mass spec experiments were already published as mentioned in the manuscript (Saha et al., 2022, Cell Reports, PMID: 35830799).